# HYPRL: Reinforcement Learning of Control Policies for Hyperproperties

**Tzu-Han Hsu**[*]   **Arshia Rafieioskouei**[*]   **Borzoo Bonakdarpour**
Michigan State University
{tzuhan, rafieios, borzoo}@msu.edu

## Abstract

Reward shaping in multi-agent reinforcement learning (MARL) for complex tasks remains a significant challenge. Existing approaches often fail to find optimal solutions or cannot efficiently handle such tasks. We propose HYPRL, a specification-guided reinforcement learning framework that learns control policies w.r.t. *hyperproperties* expressed in HyperLTL. Hyperproperties constitute a powerful formalism for specifying objectives and constraints over sets of execution traces across agents. To learn policies that maximize the satisfaction of a HyperLTL formula $\varphi$, we apply Skolemization to manage quantifier alternations and define quantitative robustness functions to shape rewards over execution traces of a Markov decision process with unknown transitions. A suitable RL algorithm is then used to learn policies that collectively maximize the expected reward and, consequently, increase the probability of satisfying $\varphi$. We evaluate HYPRL on a diverse set of benchmarks, including safety-aware planning, Deep Sea Treasure, and the Post Correspondence Problem. We also compare with specification-driven baselines to demonstrate the effectiveness and efficiency of HYPRL.

## 1 Introduction

Designing reward functions that accurately capture desirable behaviors in multi-agent reinforcement learning (MARL) remains a notorious stumbling block. Consider a wildfire scenario in a $3 \times 3$ grid-world with cells labeled $\{a, b, \ldots, i\}$ (see Figure 1). Locations $\{i, f, c\}$ are on fire, and two victims are located at $\{f, g\}$. Now, two autonomous agents are deployed from $\{a\}$ with two objectives; $\boldsymbol{O_1}$: the firefighter agent (FF) must extinguish all fire zones, and $\boldsymbol{O_2}$: the medical agent (Med) aims to rescue all victims. The agents also have to satisfy two constraints; $\boldsymbol{C_1}$: they must always remain within a 2-cell communication range, and $\boldsymbol{C_2}$: Med cannot pass any fire zone before FF extinguished the fire in that zone. The goal is to learn optimal policies for FF and Med that maximize the probability of satisfying all above requirements, as shown by the agent paths in Figure 1a.

Now, suppose we use an existing RL approach by assigning the following rewards; extinguish fire: $+50$, rescue a victim: $+10$, agent out of range: $-100$, and Med in fire zone: $-100$. These approaches would guide FF to complete $O_1$ optimally by path $a \xrightarrow{\text{R}} b \xrightarrow{\text{R}} c \xrightarrow{\text{U}} f \xrightarrow{\text{U}} i$, but due to $C_1$, it forces Med to delay the rescue of the victim in $\{g\}$ with redundant moves: $a \xrightarrow{\text{U}} d \xrightarrow{\text{D}} a \xrightarrow{\text{U}} d \xrightarrow{\text{U}} g \cdots$ (see Figure 1b). Furthermore, the victim in $\{f\}$ is trapped in fire, so satisfying $O_2$ depends on the progress of $O_1$, while respecting $C_1$ (safety) and $C_2$ (dependency). Such complex requirements make reward design in MARL particularly challenging, as policies must account for both individual objectives and relational constraints between agents with dependencies. In fact, existing RL approaches often fail to find optimal policies because agent paths are implicitly *universally* quantified, which is overly restrictive and limits the ability of RL to explore more optimal solutions. That is, the set of requirements is of the form $\forall \tau . \forall \tau' . \psi$, where $\tau$ and $\tau'$ are agents' paths and $\psi$ specifies constrains and objectives.

---

[*]Equal contribution.

39th Conference on Neural Information Processing Systems (NeurIPS 2025).

◇ ($O_1$): FF extinguishes all fires.

◇ ($O_2$): Med rescues all victims.

◇ ($C_1$): FF and Med always stay within 2-cells of each other.

◇ ($C_2$): Med must not passing any fire zone at all time.

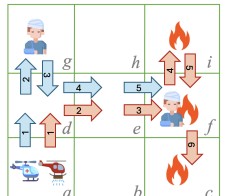 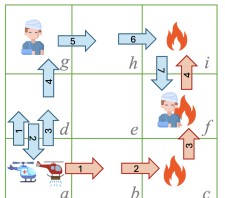 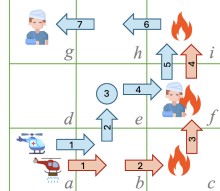

(a) Optimal policies.  (b) Med paces in $a - d$.  (c) Med pauses in $e$.

**Figure 1:** A wildfire scenario with two objectives ($O_1$, $O_2$) and two relational constraints ($C_1$, $C_2$).

Temporal logics [37] have shown to be both expressive and effective to address objectives and constraints in single-agent settings [32, 24, 2, 10, 14, 21, 22, 28, 45, 47]. However, these methods cannot handle our motivating example as it involves multiple agents. Recent works have been extended to MARL [33, 31, 20, 16], but these approaches either fail to capture relational dependencies between agents or can handle only a subset of temporal properties, such as *co-safty* [29, 41]. Moreover, specifications in temporal logics that formalize the behavior of individual traces (e.g., LTL) are again inherently universally quantified and, hence, too restrictive.

In this paper, we propose a specification-guided RL framework that leverages the powerful framework of *hyperproperties* [12] and, in particular, the temporal logic HyperLTL [13], as its formal foundation. Hyperproperties characterize requirements over sets of execution traces, allowing the specification of relational and dependent behaviors that involve multiple agents or joint system executions, rather than individual behaviors of agents. This expressiveness is particularly well-suited for capturing relational, quantified, and multi-agent objectives. Building on this foundation, we propose **Hyp**erproperties for **R**einforcement **L**earning (HYPRL), a framework that learns a collection of control policies which maximizes the probability of satisfying a hyperproperty expressed as a HyperLTL formula. In our motivating example, HyperLTL allows a specification of the form $\forall \tau. \exists \tau'. \psi$, which enables HYPRL to search for more optimal policies and identify the one in Figure 1a. The main challenge here is to deal with HyperLTL formulas with quantifier alternation, which has not been studied prior to HYPRL. Our work represents a novel step towards the automated synthesis of reward mechanisms that capture complex objectives and constraints in multi-agent setting.

**Contributions:**

1. We formulate the control policies synthesis problem for a multi-agent system as a learning problem, where the goal is to maximize the probability of satisfying a HyperLTL formula that expresses a set of objectives and constraints (Section 4).

2. We address the challenge of reasoning about quantifier alternation in a HyperLTL formula due to expressing dependencies among agents by Skolemization (Section 5.1).

3. We introduce quantitative semantics for HyperLTL that compute robustness values as rewards, providing a solution of reward shaping for optimizing hyperproperty satisfaction (Section 5.2). Finally, we use off-the-shelf RL algorithms to learn an optimal collection of policies (Section 5.3).

4. We evaluate HYPRL on a diverse set of learning tasks, including safety-aware multi-objective planning, the Post Correspondence Problem, the famous Deep Sea Treasure benchmark, and the wildfire scenario. Our results show that HYPRL is more efficient and effective in handling complex requirements compared to selected baselines (Section 6).

## 2 Related Work

**Logic-based Single-agent RL.** DIRL [24] is a framework that synthesizes an optimal policy which the specification is expressed in the language SPECTRL [23]. However, SPECTRL does not capture the full expressiveness of temporal logics as it only supports a fragment of LTL. In [32], the authors introduce *truncated* LTL for quantitative reasoning over LTL formulas, but it is limited to single-agent task specifications (similar reasonings are also presented in [44, 2, 10]). In [28], LTL formulas are encoded by replacing operators and predicates with recurrent neural networks connected according to the parse tree, forming a structure that captures the formula's semantics. In [14, 21, 45, 47, 30], the authors compose an MDP with a Limit Deterministic Büchi Automaton (LDBA), which represents the LTL specification, and solve the compositional model as a control problem.

**Logic-based Multi-Agent Reinforcement Learning.** In [25], the authors extend the SPECTRL language [23] to non-cooperative multi-agent systems and train joint policies that form a Nash Equilibrium, which requires relational reasoning. However, their algorithm follows an "enumerate-and-verify" approach by exhaustively searching RL policies to identify one that has higher probability to form such equilibrium, which leads to substantial learning overhead as the number of agents increases. Moreover, the extended language in [25] cannot handle arbitrary HyperLTL formulas. In [16], the authors propose a reward shaping method by composing an MDP with a single centralized LDBA derived from a global LTL specification. However, this approach cannot capture inter-agent dependencies, and the product construction of LDBA may suffer from state-space explosion. More recent efforts in [33, 31, 20] extend quantitative reasoning to formulas with LTL operators in a multi-agent setting. However, in contrast to HYPRL, they only cover *co-safety* specifications and do not support the full logic.

**Shield Synthesis.** Shield synthesis for RL is a technique that asks an agent to propose an action in each learning step, and a *shield* (i.e., a safety guard) evaluates whether such action is safe [3, 26, 27]. In [34], the authors apply shield synthesis in a decentralized multi-agent setting, where the learning targets are specified by deterministic finite automata. However, the specifications are limited to universal (i.e., a $\forall^*.\psi$ formula) and cannot handle properties such as planning tasks that involve dependencies (which is a $\forall\exists$ hyperproperty). Furthermore, the authors in [17] proposed *factored* shielding, which can learn multiple policies by a factorization of joint state space (i.e., decomposing one shield into multiple sub-shields). However, the main contribution is the improvement on RL scalability, but the limitation on universal-only properties remains.

## 3  Preliminaries

**Markov Decision Processes (MDP).** We model our RL problem for hyperproperties using MDP $\mathcal{M} = \langle S, s^0, A, \mathbf{P}, \mathsf{AP}, L \rangle$, where $S$ is a finite set of *states*, $s^0 \in S$ is the *initial state*, and $A$ is a finite set of *actions*. The *transition probability function* $\mathbf{P} : S \times A \times S \to [0, 1]$ assigns a probability to each transition, ensuring that for any state $s$ and action $a$, the sum of outgoing probabilities satisfies $\sum_{s' \in S} \mathbf{P}(s, a, s') = 1$. Additionally, $\mathsf{AP}$ denotes a finite set of *atomic propositions*, and $L : S \to 2^{\mathsf{AP}}$ is the *labeling* function. For example, Figure 2 represents the MDP of Figure 1, where each state can be represented by $\langle x, y \rangle$ with $x, y \in \{0, 1, 2\}$, labeled by $\{a, b, ..., i\}$, and actions $A = \{U, D, L, R\}$ (i.e., four moving directions). A *path* in $\mathcal{M}$ is a sequence $\zeta = s_0 \xrightarrow{a_0} s_1 \xrightarrow{a_1} s_2 \xrightarrow{a_2} \cdots$, where $s_i \in S$ and $a_i \in A$ for all $i \geq 0$. A *sub-path* $\zeta_{[\ell:k]}$ is a segment $s_\ell \xrightarrow{a_\ell} \cdots \xrightarrow{a_{k-1}} s_k$, for $0 \leq \ell < k < |\zeta|$. We write

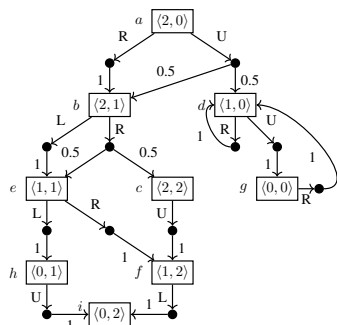

**Figure 2:** The MDP of The grid-world in Figure 1.

$\mathcal{Z}^*$ and $\mathcal{Z}^\omega$ to denote the sets of all finite and infinite paths, respectively. A (deterministic) *policy* $\pi : \mathcal{Z}^* \to A$ maps each finite path to a (fixed) action $a \in A$. The *trace* $t$ of a path $\zeta$ is the sequence of labels $\mathsf{Tr}(\zeta) = t(0)t(1)t(2)\cdots$, where $t(i) = L(s_i)$ for all $i \geq 0$. Slightly abusing notation, we use $\mathsf{Traces}(\mathcal{Z}^*)$ and $\mathsf{Traces}(\mathcal{Z}^\omega)$ to denote the sets of all finite and infinite traces, respectively.

**Finite Semantics for HyperLTL.** The syntax of HyperLTL [13] is given by the following grammar:

$$\varphi ::= \exists\tau.\varphi \mid \forall\tau.\varphi \mid \psi \qquad \psi ::= p_\tau \mid \neg\psi \mid \psi \vee \psi \mid \bigcirc\psi \mid \psi\,\mathcal{U}\,\psi,$$

where $p \in \mathsf{AP}$ is an atomic proposition, $\tau$ is a *trace variable*, and the second rule (for unquantified formulas) produces LTL formulas. The Boolean connectives $\neg$ and $\vee$ have their standard meanings, while $\bigcirc$ and $\mathcal{U}$ denote the "next" and "until" temporal operators, respectively. Other Boolean and temporal operators are derived as syntactic sugar: true $\triangleq p_\tau \vee \neg p_\tau$, false $\triangleq \neg$true, $\psi_1 \to \psi_2 \triangleq \neg\psi_1 \vee \psi_2$, $\Diamond\psi \triangleq$ true $\mathcal{U}\,\psi$, and $\Box\psi \triangleq \neg\Diamond\neg\psi$, where '$\Diamond$' and '$\Box$' are the *eventually* and *always* temporal operators. In a quantified formula, $\exists\tau$ means "along some trace $\tau$", and $\forall\tau$ means "along all traces $\tau$". We write $Vars(\varphi)$ for the set of trace variables in a formula $\varphi$. A formula is *closed* if all $\tau \in Vars(\varphi)$ are quantified, with no variable quantified twice. Since RL algorithms operate on finite samples, we adopt the *finite semantics* of HyperLTL [11] over finite trace assignments, where a partial mapping $\Pi : Vars(\varphi) \rightharpoonup (2^{\mathsf{AP}})^*$ assigns each $\tau \in Vars(\varphi)$ to a finite trace. Given $\Pi$, a trace variable $\tau$, and a finite trace $t \in (2^{\mathsf{AP}})^*$, we write $\Pi[\tau \to t]$ for the assignment identical to $\Pi$, except that $\tau$ is mapped to $t$. We denote by $\Pi_\emptyset$ the empty trace assignment. For $t \in \Pi$, we refer to traces in the image of $\Pi$. Throughout this paper, we abbreviate tuples $\langle x_1, \ldots, x_n \rangle$ as $\langle x_i \rangle_{i \in 1,\ldots,n}$. An *interpretation* of

a HyperLTL formula $\varphi = \mathbb{Q}_1\tau_1.\ldots.\mathbb{Q}_n\tau_n.\,\psi$, denoted by $\mathcal{T} = \langle T_{\tau_i}\rangle_{i\in\{1,\ldots,|Vars(\varphi)|\}}$, is a tuple of sets of traces, where we have one set $T_{\tau_i}$ per trace variable $\tau_i$, denoting the set of traces that can be assigned to $\tau_i$. Let $\mathcal{D}_{\pi_i}$ be the distribution over a set of paths induced by a policy $\pi_i$, and we write $\mathcal{Z}_{\tau_i} \sim \mathcal{D}_{\pi_i}$ to denote a set of paths $\mathcal{Z}_{\tau_i}$ sampled from $\mathcal{D}_{\pi_i}$, such that $\pi_i$ is the policy associated with the trace variable $\tau_i$, for each $i \in \{1,\ldots,|Vars(\varphi)|\}$ (each trace variable ranges over the possible behaviors of an agent). We also define a *family* of sampled sets $\mathcal{S} = \langle \mathcal{Z}_{\tau_i} \sim \mathcal{D}_{\pi_i}\rangle_{i\in\{1,\ldots,|Vars(\varphi)|\}}$. That is, for each $\tau_i$ in $Vars(\varphi)$, $T_{\tau_i} = \mathsf{Traces}(\mathcal{Z}_{\tau_i} \sim \mathcal{D}_{\pi_i})$ is the set of traces that $\tau_i$ can range over, which comes from the sampled paths from the associated policy $\pi_i$. Abusing notation, we write $\mathcal{T} = \mathsf{Traces}(\mathcal{S})$ as the tuple of sets of sampled traces. The satisfaction relation $\models$ maps a formula $\varphi$ to a model $(\mathcal{T},\Pi,i)$, where $i \in \mathbb{Z}_{\geq 0}$ indicates the current evaluation position. Formally:

$$
\begin{array}{llll}
(\mathcal{T},\Pi,0) & \models \exists\tau.\,\psi & \text{iff} & \text{there is a } t \in T_\tau, \text{ such that } (\mathcal{T},\Pi[\tau \to t],0) \models \psi \\
(\mathcal{T},\Pi,0) & \models \forall\tau.\,\psi & \text{iff} & \text{for all } t \in T_\tau, \text{ such that } (\mathcal{T},\Pi[\tau \to t],0) \models \psi \\
(\mathcal{T},\Pi,i) & \models p_\tau & \text{iff} & p \in \Pi(\tau)(i) \\
(\mathcal{T},\Pi,i) & \models \neg\psi & \text{iff} & (\mathcal{T},\Pi,i) \not\models \psi \\
(\mathcal{T},\Pi,i) & \models \psi_1 \vee \psi_2 & \text{iff} & (\mathcal{T},\Pi,i) \models \psi_1 \text{ or } (\mathcal{T},\Pi,i) \models \psi_2 \\
(\mathcal{T},\Pi,i) & \models \bigcirc\psi & \text{iff} & (\mathcal{T},\Pi,i+1) \models \psi \text{ and for all } t \in \Pi.|t| \geq i+1 \\
(\mathcal{T},\Pi,i) & \models \psi_1\,\mathcal{U}\,\psi_2 & \text{iff} & \text{there exists } j \geq i \text{ with } j < \min_{t\in\Pi}|t|, \text{ such that } (\mathcal{T},\Pi,j) \models \psi_2 \\
& & & \text{and for all } k \in [i,j),(\mathcal{T},\Pi,k) \models \psi_1.
\end{array}
$$

We say that an interpretation $\mathcal{T}$ satisfies a HyperLTL formula $\varphi$, written as $\mathcal{T} \models \varphi$, if $(\mathcal{T},\Pi_\emptyset,0) \models \varphi$. Likewise, a family of samples $\mathcal{S}$ (induced by each $\pi_{\tau_i}$ associated with each $\tau_i \in Vars(\varphi)$), satisfies a sentence $\varphi$ if $\langle\mathsf{Traces}(\mathcal{Z}_{\tau_i} \sim \mathcal{D}_{\pi_i})\rangle_{i\in\{1,\ldots,|Vars(\varphi)|\}} \models \varphi$.

**Example.**  The following HyperLTL formula captures the objectives and constraints of our motivating example described in Figure 1, where $\tau_1$ is the path for FF and $\tau_2$ is the path for Med:

$$Specification:\ \varphi_{\mathsf{Rescue}} \triangleq \forall\tau_1.\exists\tau_2.(\psi_{\mathsf{fire}} \wedge \psi_{\mathsf{save}} \wedge \psi_{\mathsf{dist}} \wedge \psi_{\mathsf{safe}})$$

$$O_1:\ \psi_{\mathsf{fire}} \triangleq \Diamond(i_{\tau_1}) \wedge \Diamond(f_{\tau_1}) \wedge \Diamond(c_{\tau_1}) \qquad C_1:\ \psi_{\mathsf{dist}} \triangleq \Box(|\mathsf{Location}_{\tau_1} - \mathsf{Location}_{\tau_2}| < 3)$$

$$O_2:\ \psi_{\mathsf{save}} \triangleq \Diamond(g_{\tau_2}) \wedge \Diamond(f_{\tau_2}) \qquad C_2:\ \psi_{\mathsf{safe}} \triangleq (\neg i_{\tau_2}\,\mathcal{U}\,i_{\tau_1}) \wedge (\neg f_{\tau_2}\,\mathcal{U}\,f_{\tau_1}) \wedge (\neg c_{\tau_2}\,\mathcal{U}\,c_{\tau_1})$$

For instance, the dependency constraint $C_2$ for "Med cannot enter any fire zone until FF extinguished the fire in that zone" is expressed using the conjunction of temporal *until* operators. Notice that $\varphi_{\mathsf{Rescue}}$ features $\forall\exists$ quantifier alternation, which increases the complexity of reasoning about hyperproperties [8]. We emphasize that most RL approaches assume purely universal forms (i.e., $\forall^*$), which cannot capture agent dependencies and often yield sub-optimal solutions. For instance, if FF ignores that Med must wait until the fire is extinguished to rescue a victim, FF may follow its own optimal path that causes unnecessary delay for Med (see Figure 1c).

## 4   Problem Statement

Let us use $\star$ to denote optimality (e.g., $\pi^\star$ denotes an optimal policy). The following optimization problem formulates policy synthesis as a learning problem.

> Given an MDP $\mathcal{M}$ with unknown transitions and a HyperLTL formula $\varphi$ of the form $\mathbb{Q}_1\tau_1.\ldots.\mathbb{Q}_n\tau_n.\,\psi$, our goal is to identify a tuple of $n$ policies $\langle\pi_1^\star,\ldots,\pi_n^\star\rangle$, such that:
>
> $$\langle\pi_i^\star\rangle_{i\in\{1,\ldots,n\}} \in \left[\arg\max_{\langle\pi_i\rangle} \mathbb{P}\Big[\langle\mathsf{Traces}(\mathcal{Z}_{\tau_i} \sim \mathcal{D}_{\pi_i})\rangle \models \varphi\Big]\right]_{i\in\{1,\ldots,n\}}$$
>
> where $\mathcal{D}_{\pi_1},\ldots,\mathcal{D}_{\pi_n}$ are the distributions over set of paths generated by policy spaces $\pi_1,\ldots,\pi_n$. That is, $\langle\pi_1^\star,\ldots,\pi_n^\star\rangle$ maximizes the probability $\mathbb{P}$ such that the generated tuple of sets of traces $\langle\mathsf{Traces}(\mathcal{Z}_{\tau_1} \sim \mathcal{D}_{\pi_1}),\ldots,\mathsf{Traces}(\mathcal{Z}_{\tau_n} \sim \mathcal{D}_{\pi_n})\rangle$ from $\mathcal{M}$ satisfies $\varphi$.

**Example.**  Consider the MDP in Figure 2 and the following HyperLTL formula:

$$\varphi_{\mathsf{exp}} \triangleq \forall\tau_1.\exists\tau_2.\Big(\Diamond i_{\tau_1} \wedge \Box\,dist(\langle x,y\rangle_{\tau_1},\langle x,y\rangle_{\tau_2}) < 3\Big)$$

Suppose agent FF with $\pi_1$ draws samples $\mathcal{Z}_{\tau_1} = \{\zeta_{\mathsf{FF}}^1, \zeta_{\mathsf{FF}}^2\}$ from the MDP:

$$\zeta_{\mathsf{FF}}^1 : \underbrace{\langle 2,0\rangle}_{a} \xrightarrow{R} \underbrace{\langle 2,1\rangle}_{b} \xrightarrow{R} \underbrace{\langle 2,2\rangle}_{c} \xrightarrow{U} \underbrace{\langle 1,2\rangle}_{f} \xrightarrow{L} \underbrace{\langle 0,2\rangle}_{i} \qquad \zeta_{\mathsf{FF}}^2 : \underbrace{\langle 2,0\rangle}_{a} \xrightarrow{R} \underbrace{\langle 2,1\rangle}_{b} \xrightarrow{R} \underbrace{\langle 1,1\rangle}_{e} \xrightarrow{L} \underbrace{\langle 0,1\rangle}_{h} \xrightarrow{U} \underbrace{\langle 0,2\rangle}_{i}$$

Agent Med with policy $\pi_2$ draws $\mathcal{Z}_{\tau_2} = \{\zeta_{\mathsf{Med}}^1, \zeta_{\mathsf{Med}}^2\}$:

$$\zeta_{\mathsf{Med}}^1 : \underbrace{\langle 2, 0\rangle}_{a} \xrightarrow{\mathrm{U}} \underbrace{\langle 1, 0\rangle}_{d} \xrightarrow{\mathrm{U}} \underbrace{\langle 0, 0\rangle}_{g} \xrightarrow{\mathrm{R}} \underbrace{\langle 1, 0\rangle}_{d} \xrightarrow{\mathrm{R}} \underbrace{\langle 1, 0\rangle}_{d} \qquad \zeta_{\mathsf{Med}}^2 : \underbrace{\langle 2, 0\rangle}_{a} \xrightarrow{\mathrm{U}} \underbrace{\langle 2, 1\rangle}_{b} \xrightarrow{\mathrm{L}} \underbrace{\langle 1, 1\rangle}_{e} \xrightarrow{\mathrm{R}} \underbrace{\langle 1, 2\rangle}_{f} \xrightarrow{\mathrm{L}} \underbrace{\langle 0, 2\rangle}_{i}$$

Notice that, the number of samples can be more than two. We now calculate the probability of satisfying $\varphi$ using $\mathcal{Z}_{\tau_1}$ and $\mathcal{Z}_{\tau_2}$ (obtained by $\pi_1$ and $\pi_2$) as follows:

$$\mathsf{Traces}(\langle \{\zeta_{\mathsf{FF}}^1\}, \mathcal{Z}_{\tau_2}\rangle) \models \varphi_{\exp} \quad \mathsf{Traces}(\langle \{\zeta_{\mathsf{FF}}^2\}, \mathcal{Z}_{\tau_2}\rangle) \models \varphi_{\exp},$$

where $\zeta_{\mathsf{Med}}^2$ is the witness to $\tau_2$ (existentially quantified) for both satisfaction relations. Hence, given $\mathcal{Z}_1$ and $\mathcal{Z}_2$, the probability of satisfying $\varphi_{\exp}$ is evaluated as:

$$\mathbb{P}_{\langle \pi_1, \pi_2\rangle}\Big[\mathsf{Traces}(\langle \mathcal{Z}_{\tau_1}, \mathcal{Z}_{\tau_2}\rangle) \models \varphi_{\exp}\Big] = 1$$

Now, if $\varphi_{\exp}$ had the form $\forall\forall$, the evaluation has to go over all combinations between $\mathcal{Z}_{\tau_1}$ and $\mathcal{Z}_{\tau_2}$:

$$\mathsf{Traces}(\langle \{\zeta_{\mathsf{FF}}^1\}, \{\zeta_{\mathsf{Med}}^1\}\rangle) \not\models \varphi_{\exp} \quad \mathsf{Traces}(\langle \{\zeta_{\mathsf{FF}}^1\}, \{\zeta_{\mathsf{Med}}^2\}\rangle) \models \varphi_{\exp}$$

$$\mathsf{Traces}(\langle \{\zeta_{\mathsf{FF}}^2\}, \{\zeta_{\mathsf{Med}}^1\}\rangle) \not\models \varphi_{\exp} \qquad \mathsf{Traces}(\langle \{\zeta_{\mathsf{FF}}^2\}, \{\zeta_{\mathsf{Med}}^2\}\rangle) \models \varphi_{\exp}$$

Thus, the satisfaction probability of $\forall\forall\psi$ would be 0.5. This example demonstrates that the probability of satisfying a HyperLTL formula crucially depends on its quantifier structure.

# 5 Algorithmic Details of HYPRL

To solve the problem formally stated in Section 3, our algorithm proceeds in the following three steps. We first Skolemize $\varphi$ [40] to eliminate quantifier alternations and simplify the learning task. We then define quantitative semantics for HyperLTL, converting satisfaction checking into robustness value optimization. Finally, we train a neural network using these robustness signals to learn optimal policies that solve the original learning problem.

## 5.1 Step 1: HyperLTL Skolemization

Let a HyperLTL formula be of the form $\varphi = \mathbb{Q}_1\tau_1.\mathbb{Q}_2\tau_2.\ldots.\mathbb{Q}_n.\tau_n.\ \psi(\tau_1, \tau_2, \ldots, \tau_n)$, where for $1 \leq \ell \leq n$, each $\mathbb{Q}_\ell \in \{\forall, \exists\}$ quantifies a trace variable $\tau_\ell$, and $\psi$ is a quantifier-free LTL formula. We first Skolemize $\varphi$, producing $\mathbf{Skolem}(\varphi)$, to eliminate quantifier alternations. We define $\mathbb{Q}^\exists = \{i \mid \mathbb{Q}_i = \exists\}$ and $\mathbb{Q}^\forall = \{j \mid \mathbb{Q}_j = \forall\}$ as the sets of existential and universal quantifier indices, respectively. For each $i \in \mathbb{Q}^\exists$, we denote $\mathbb{Q}_i^\forall = \{j < i \mid \mathbb{Q}_j = \forall\}$ as the index set of all universal quantifiers preceding $\mathbb{Q}_i$. A *Skolem function* for each $i \in \mathbb{Q}^\exists$ is defined as $\mathbf{f}_i : \mathcal{T}^{|\mathbb{Q}_i^\forall|} \to \mathcal{T}$, and reduces to a constant function when $\mathbb{Q}_i^\forall = \emptyset$. A trace assignment $\Pi$ is *consistent* with $\mathbf{f}_i$, if $\Pi(\tau_{i_j}) \in \mathcal{T}$ for all $j \in \mathbb{Q}_i^\forall$, and $\Pi(\tau_i) = \mathbf{f}_i\big(\Pi(\tau_{i_1}), \Pi(\tau_{i_2}), \ldots, \Pi(\tau_{i_{|\mathbb{Q}_i^\forall|}})\big)$ for all $i \in \mathbb{Q}^\exists$, where $\mathbb{Q}_i^\forall = \{i_1 < i_2 < \cdots < i_{|\mathbb{Q}_i^\forall|}\}$. If $(\mathcal{T}, \Pi, 0) \models \varphi$ for every trace assignment $\Pi$ consistent with all $\mathbf{f}_i$, then each $\mathbf{f}_i$ is said to *witness* the satisfaction of $\varphi$ [43]. For the inner LTL formula $\psi$ (i.e., obtaining $\mathbf{Skolem}(\psi)$), we replace each proposition $p_{\tau_i}$ with $p_{\mathbf{f}_i}$ for all $p \in \mathsf{AP}$ and $i \in \mathbb{Q}^\exists$, thereby instantiating variables of the existentially quantified paths via their Skolem witnesses. In general, a Skolemized $\varphi$ is of the following form:

$$\mathbf{Skolem}(\varphi) = \underbrace{\exists \mathbf{f}_i(\tau_{i_1}, \ldots, \tau_{i_{|\mathbb{Q}_i^\forall|}})}_{\text{for each } i \in \mathbb{Q}^\exists}.\quad \underbrace{\forall\tau_j.}_{\text{for each } j \in \mathbb{Q}^\forall}\quad \mathbf{Skolem}(\psi) \tag{1}$$

Based on this transformation, we re-write the problem statement from Section 3 as follows. We first define the *image* of $\mathbf{f}_i$:

$$Img(\mathbf{f}_i) \triangleq \{\mathbf{f}_i(t_{i_1}, \ldots, t_{i_{|\mathbb{Q}_i^\forall|}}) \mid t_{i_j} \in \mathsf{Traces}(\mathcal{Z}_{\tau_{i_j}} \sim \mathcal{D}_{\pi_{i_j}}),\ j \in \mathbb{Q}_i^\forall\}$$

That is, $Img(\mathbf{f}_i)$ is the set of mapped traces (which $\tau_{i_j}$ for each $i \in \mathbb{Q}^\exists$ ranges over) from all possible preceding $\forall$-quantified $\tau_{i_j}$, where each $t_{i_j}$ is from its own sampled trace set $\mathcal{Z}_{\tau_{i_j}}$. Now, let us use $\bowtie$ as a notation to ensure the collection of trace sets are ordered w.r.t. their path indices. Given two tuples of sets of traces $\mathcal{T}_1$ and $\mathcal{T}_2$, we define $\mathcal{T}_1 \bowtie \mathcal{T}_2 \triangleq \langle \mathsf{Traces}(\mathcal{Z}_{\tau_x})\rangle_{x \in \{1\cdots n\}}$, where each

$$\begin{aligned}
\rho\big(\mathsf{Tr}(\zeta_{[\ell:k]}),\psi\big) &= \rho_{min} \text{ if } \mathsf{Tr}(\zeta_{[\ell:..]}) = \epsilon \text{ and } \rho\big(\mathsf{Tr}(\zeta_{[\ell:k]}),\psi\big) \text{ otherwise.} \\
\rho\big(\mathsf{Tr}(\zeta_{[\ell:k]}),\mathsf{true}\big) &= \rho_{max} \\
\rho\big(\mathsf{Tr}(\zeta_{[\ell:k]}),f\big(L(s_\ell)<c\big)\big) &= c - f\big(L(s_\ell)\big) \\
\rho\big(\mathsf{Tr}(\zeta_{[\ell:k]}),\neg\psi\big) &= -\rho\big(\mathsf{Tr}(\zeta_{[\ell:k]}),\psi\big) \\
\rho\big(\mathsf{Tr}(\zeta_{[\ell:k]}),\bigcirc\psi\big) &= \rho\big(\mathsf{Tr}(\zeta_{[\ell+1:k]}),\psi\big) \text{ if } (k>\ell). \\
\rho\big(\mathsf{Tr}(\zeta_{[\ell:k]}),\square\psi\big) &= \min_{i\in[\ell,k)}\rho\big(\mathsf{Tr}(\zeta_{[i:k]}),\psi\big) \\
\rho\big(\mathsf{Tr}(\zeta_{[\ell:k]}),\Diamond\psi\big) &= \max_{i\in[\ell,k)}\rho\big(\mathsf{Tr}(\zeta_{[i:k]})\psi\big) \\
\rho\big(\mathsf{Tr}(\zeta_{[\ell:k]}),\psi_1\wedge\psi_2\big) &= \min\big(\rho\big(\mathsf{Tr}(\zeta_{[\ell:k]}),\psi_1\big)\rho\big(\mathsf{Tr}(\zeta_{[\ell:k]}),\psi_2\big)\big) \\
\rho\big(\mathsf{Tr}(\zeta_{[\ell:k]}),\psi_1\vee\psi_2\big) &= \max\big(\rho\big(\mathsf{Tr}(\zeta_{[\ell:k]}),\psi_1\big)\rho\big(\mathsf{Tr}(\zeta_{[\ell:k]}),\psi_2\big)\big) \\
\rho\big(\mathsf{Tr}(\zeta_{[\ell:k]}),\psi_1\,\mathcal{U}\,\psi_2\big) &= \max_{i\in[\ell,k)}\Big(\min\big(\rho\big(\mathsf{Tr}(\zeta_{[i:k]}),\psi_2\big),\min_{j\in[\ell,i)}\rho\big(\mathsf{Tr}(\zeta_{[j:i]}),\psi_1\big)\big)\Big)
\end{aligned}$$

**Figure 3:** Quantitative semantics for LTL.

$\mathsf{Traces}(\mathcal{Z}_{\tau_x})$ is either in $\mathcal{T}_1$ or $\mathcal{T}_2$ (and not both). Given an MDP $\mathcal{M}$ and a HyperLTL specification $\varphi$ of the form $\mathbb{Q}_1\tau_1.\mathbb{Q}_2\tau_2.\ldots.\mathbb{Q}_n\tau_n.\ \psi$, our goal is to compute (1) a tuple of Skolem witnesses $\langle\mathbf{f}_i\rangle_{i\in\mathbb{Q}^\exists}$, and (2) a tuple of policies $\langle\pi_j^\star\rangle_{j\in\mathbb{Q}^\forall}$, such that:

$$\langle\pi_j^\star\rangle_{j\in\mathbb{Q}^\forall}\in\left[\underset{\langle\pi_j\rangle}{\arg\max}\ \mathbb{P}\Big[\langle Img(\mathbf{f}_i)\rangle\bowtie\langle\mathsf{Traces}(\mathcal{Z}_{\tau_j}\sim\mathcal{D}_{\pi_j})\rangle\models\mathbf{Skolem}(\varphi)\Big]\right]_{i\in\mathbb{Q}^\exists,j\in\mathbb{Q}^\forall}$$

That is, the tuple of policies $\langle\pi_j^\star\rangle$ maximizes the probability that the ordered collection of (1) the generated traces of all universal quantifiers $\langle\mathsf{Traces}(\mathcal{Z}_{\tau_j}\sim\mathcal{D}_{\pi_j})\rangle_{j\in\mathbb{Q}^\forall}$ and (2) the sets of mapped traces (i.e., the image) of each Skolem witness for all existential quantifiers $\langle Img(\mathbf{f}_i)\rangle_{i\in\mathbb{Q}^\exists}$ together satisfies $\mathbf{Skolem}(\varphi)$. Notice that in the updated problem statement, we compute policies only for universally quantified traces, while Skolem functions are learned for existentially quantified traces to witness the optimality.

### 5.2 Step 2: Policy Learning with Quantitative Semantics

To transform the satisfaction checking problem (i.e., determining $\models$) into an optimization task, we define quantitative semantics for HyperLTL, extended from [32]. In particular, we evaluate the Skolemized HyperLTL formula $\mathbf{Skolem}(\varphi)$ over tuples of sampled paths $\langle\zeta_1,\zeta_2,\ldots,\zeta_n\rangle$ from $\mathcal{M}$.

**Robustness for a Single Trace.** Let $\mathbb{R}$ be the set of real numbers and $\Psi$ the set of all LTL formulas. We define a valuation function $f:2^{\mathsf{AP}}\to\mathbb{R}$ that assigns a real value to a set of atomic propositions, provided as part of the input. Given a state $s\in S$ of an MDP $\mathcal{M}$, the quantitative semantics are defined over predicates in the form of $f\big(L(s)\big)<c$, where $c$ is a user-specified threshold. Next, we define a *robustness function* $\rho:\mathsf{Traces}(\mathcal{Z}^*)\times\Psi\to\mathbb{R}$ that assigns a robustness value to a finite trace for an LTL formula. Intuitively, the robustness value evaluates "how far" the given finite trace is from satisfying $\psi$. The complete quantitative semantics is shown in Figure 3. We use constants $\rho_{max}$ and $\rho_{min}$ for the maximum and minimum robustness values, respectively. Given a trace, a higher $\rho$ value implies the trace has higher robustness to satisfy $\psi$, and a lower $\rho$ value means the trace is less likely to satisfy $\psi$ (e.g., a potential violation).

Formally, given an LTL formula $\psi$ and an MDP $\mathcal{M}$, a path $\zeta$ with a higher robustness value indicates that it has higher probability to satisfy $\psi$. The optimization task of seeking a single policy $\pi^\star$ is:

$$\pi^\star\in\underset{\pi}{\arg\max}\ \underset{\zeta\sim\mathcal{D}_\pi}{\mathbb{P}}\Big[\rho\big(\mathsf{Tr}(\zeta_{[0:k]}),\psi\big)\overset{\star}{\to}\rho_{max}\Big]$$

Here, for simplicity, we use the notation $\overset{\star}{\to}$ to represent convergence. That is, $\pi^\star$ maximizes the probability of satisfying $\psi$ over the distribution of paths generated by policy $\pi$.

**Robustness for a Tuple of Traces.** To evaluate the robustness value over multiple traces, we first define a $\mathsf{zip}$ function that pointwise bundles a tuple of traces. Given a tuple of finite traces $\langle t_1,\ldots,t_n\rangle$, we derive a zipped trace $\mathsf{zip}(\langle t_1,\ldots,t_n\rangle)$ and for all $i\geq 0$, $\mathsf{zip}(\langle t_1,\ldots,t_n\rangle)(i)\triangleq\langle t_1(i),\ldots,t_n(i)\rangle$. Given an LTL formula, a tuple of paths $\langle\zeta_1,\zeta_2,\ldots,\zeta_n\rangle$ has higher probability of satisfying $\psi$ if the

robustness value of $\mathsf{zip}\big(\langle\mathsf{Tr}(\zeta_{1[0:k_1]}),\mathsf{Tr}(\zeta_{2[0:k_2]}),\ldots,\mathsf{Tr}(\zeta_{n[0:k_n]})\rangle\big)$ converges to $\rho_{max}$ w.r.t. $\psi$ for some $k_1,\ldots,k_n$, where $0\leq k_\ell\leq|\zeta_\ell|$ for each $1\leq\ell\leq n$. Thus, the optimization task of computing a tuple of policies $\langle\pi_1^\star,\pi_2^\star,\ldots,\pi_n^\star\rangle$ that maximizes the robustness becomes:

$$\langle\pi_\ell^\star\rangle_{\ell\in\{1,\ldots n\}}\in\left[\arg\max_{\langle\pi_\ell\rangle}\mathbb{P}\Big[\rho\big(\mathsf{zip}(\langle\mathsf{Tr}(\zeta_{\ell[0:k_\ell]}\sim\mathcal{D}_{\pi_\ell})\rangle),\psi\big)\xrightarrow{\star}\rho_{max}\Big]\right]_{\ell\in\{1,\ldots,n\}}$$

This formulation reflects that LTL formulas are implicitly universally quantified as we emphasized in Section 1. For instance, model checking $\forall^\star.\psi$ reduces to checking $\forall.\psi$ via self-composition [5, 8].

**Robustness for Skolemized HyperLTL.** Optimizing an alternating HyperLTL formula requires that policies for universally quantified paths simultaneously optimize the existentially quantified paths. In order to preserve ordering, we use $\cup_\leq$ to make sure the union of trace tuples are in order w.r.t. their path indices. That is, given two tuple of traces $\langle\cdot\rangle_1$ and $\langle\cdot\rangle_2$, we define $\langle\cdot\rangle_1\cup_\leq\langle\cdot\rangle_1\triangleq\langle\mathsf{Tr}(\zeta_x)\rangle_{x\in\{1\cdots n\}}$, where each $\mathsf{Tr}(\zeta_x)$ is either from $\langle\cdot\rangle_1$ or $\langle\cdot\rangle_2$. Given a Skolemized HyperLTL formula, the inner LTL formula is denoted as $\mathbf{Skolem}(\psi)$, as defined in (1). For each $i\in\mathbb{Q}^\exists$ and $j\in\mathbb{Q}^\forall$, we say a tuple of paths $\langle\zeta_1,\zeta_2,\ldots,\zeta_n\rangle$ satisfies $\psi$ if and only if:

$$\left[\rho\Big(\mathsf{zip}(\langle\mathsf{Tr}(\zeta_{i[0:k_i]})\rangle\cup_\leq\langle\mathsf{Tr}(\zeta_{j[0:k_j]})\rangle),\mathbf{Skolem}(\psi)\Big)\right]_{i\in\mathbb{Q}^\exists,j\in\mathbb{Q}^\forall}\xrightarrow{\star}\rho_{max}$$

Essentially, the set $\langle\mathsf{Tr}(\zeta_{i[0:k_i]})\rangle_{i\in\mathbb{Q}^\exists}$ captures the tuple of traces produced by each Skolem function for each existential quantifiers (i.e., a trace from $Img(\mathbf{f}_i)$) for each $i\in\mathbb{Q}^\exists$), while $\langle\mathsf{Tr}(\zeta_{j[0:k_j]})\rangle_{j\in\mathbb{Q}^\forall}$ corresponds to the tuple of traces for each $j\in\mathbb{Q}^\forall$. That is, we can formalize the optimization task for a Skolemized HyperLTL as follows:

$$\left[\langle\pi_i^\star\rangle\cup_\leq\langle\pi_j^\star\rangle\in\arg\max_{\langle\pi_i\rangle\cup_\leq\langle\pi_j\rangle}\mathbb{P}\Big[\rho\Big(\mathsf{zip}(\langle\mathsf{Tr}(\zeta_{i[0:k_i]}\sim\mathcal{D}_{\pi_i})\rangle\cup_\leq\langle\mathsf{Tr}(\zeta_{j[0:k_j]}\sim\mathcal{D}_{\pi_j})\rangle),\right.$$

$$\left.\mathbf{Skolem}(\psi)\Big)\xrightarrow{\star}\rho_{max}\Big]\right]_{i\in\mathbb{Q}^\exists,j\in\mathbb{Q}^\forall}\qquad(2)$$

Notice that, for each $i\in\mathbb{Q}^\exists$ (i.e., a Skolem function), the evaluation of $\rho$ dependes on whether the sampled trace $\zeta_i\sim\mathcal{D}_{\pi_i}$ *serves* as a witness for its preceding $\langle\zeta_{i_j}\rangle_{i_j\in\mathbb{Q}_i^\forall}$. That is, $\rho(\mathbf{f}_i,\psi)\triangleq\rho\big(\mathbf{f}_i(\mathsf{Tr}(\zeta_{i_1}),\ldots,\mathsf{Tr}(\zeta_{i_{|\mathbb{Q}_i^\forall|}})),\psi\big)$. Hence, the optimization of $\langle\pi_i^\star\rangle$ depends on $\langle\pi_j^\star\rangle$, which is necessary to capture the semantics of a HyperLTL formula with quantifier alternation (we provide a comprehensive example in Appendix C). Finally, we show that the set of optimal policies obtained by (2) also solves the original optimization problem in Section 3 (detailed proof in Appendix A).

> **Theorem 1** *Given an MDP $\mathcal{M}$ and a HyperLTL formula $\varphi$, an optimal tuple of policies $\langle\pi_i^\star\rangle_{i\in\mathbb{Q}^\exists}\cup_\leq\langle\pi_j^\star\rangle_{j\in\mathbb{Q}^\forall}$ for $\mathbf{Skolem}(\varphi)$ is also an optimal tuple of policies for $\varphi$, that optimizes the probability of satisfying $\varphi$ in $\mathcal{M}$ (the problem statement in Section 3).*

### 5.3 Step 3: Reinforcement Learning for HyperLTL

Now, we solve the optimization task in (2) using RL, where the reward signal is defined by the robustness values from Section 5.2. As our MDP model lacks a built-in reward function, we use robustness values to guide learning. To avoid confusion with standard RL terminology, we refer to expected robustness value when mentioning expected rewards. For simplicity, we assume all sampled paths have equal length.

**Optimizing Expected Reward.** The goal of this step is to learn an optimal neural network $\mathcal{NN}^\star$, a parameterized compositional function that synthesizes the policies solving (2). The construction of learning constraints is inspired by the well-known *Bellman Equation* [4], which characterizes the optimal expected reward of a decision process as a recursive function of immediate reward and the value of future states. Let us use $s_k$ to denoted the state of a zipped path at position $k$ (i.e., $s_k\triangleq\mathsf{zip}(\langle t_1,\ldots,t_n\rangle)(k)$). We define the *immediate reward* for a state-action pair $(s_k,a_k)$ based on the robustness value of the zipped trace from (2):

$$R(s_k,a_k)\triangleq\left[\rho\Big(\mathsf{zip}(\langle\mathsf{Tr}(\zeta_{i[0:k]})\rangle\cup_\leq\langle\mathsf{Tr}(\zeta_{j[0:k]})\rangle),\mathbf{Skolem}(\psi)\Big)\right]_{i\in\mathbb{Q}^\exists,j\in\mathbb{Q}^\forall}$$

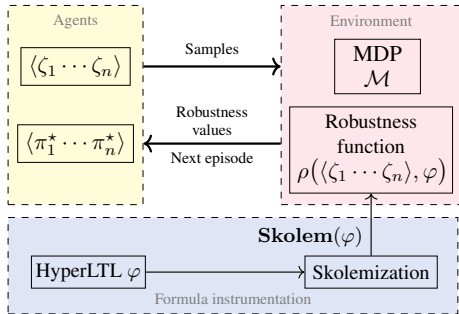

**Figure 4:** Overview of HYPRL.

**Table 1:** HyperLTL specifications of case studies.

| | |
|---|---|
| (SRL) | $\forall \tau_1. \exists \tau_2.\ \square \langle x_{\tau_1}, y_{\tau_1} \rangle \neq \langle x_{\tau_2}, y_{\tau_2} \rangle \wedge \lozenge \langle x_{\tau_1}, y_{\tau_1} \rangle = \langle x_{G1}, y_{G1} \rangle \wedge$ $\lozenge \langle x_{\tau_2}, y_{\tau_2} \rangle = \langle x_{G2}, y_{G2} \rangle$ |
| (DST) | $\forall \tau_1. \exists \tau_2.\ \lozenge (T1_{\tau_1} \wedge \lozenge (T2_{\tau_1} \wedge \lozenge (T3_{\tau_1}) \dots)) \wedge$ $\square\,(step_{\tau_2} < \delta) \wedge \square\,(|pos_{\tau_1} - pos_{\tau_2}| < 1)$ |
| (PCP) | $\forall \tau_1. \exists \tau_2.\ \psi_{\mathsf{SemiMatch}_{\tau_1}}\ \mathcal{U}\ (\psi_{\mathsf{Extend}_{\tau_1}, \tau_2} \wedge \bigwedge_{p \in \mathsf{AP}} (p_{top_{\tau_2}} \leftrightarrow p_{bot_{\tau_2}}))$ $\psi_{\mathsf{SemiMatch}_{\tau_1}} \triangleq \left[ \bigwedge_{p \in \mathsf{AP}} (p_{top_{\tau_1}} \leftrightarrow p_{bot_{\tau_1}}) \right] \mathcal{U}\ (\#_{top_{\tau_1}} \oplus \#_{bot_{\tau_1}})$ $\varphi_{\mathsf{Extend}_{\tau_1}, \tau_2} \triangleq \left[ \bigwedge_{p \in \mathsf{AP}} ((p_{top_{\tau_1}} \leftrightarrow p_{top_{\tau_2}}) \wedge (p_{bot_{\tau_1}} \leftrightarrow p_{bot_{\tau_2}})) \right] \mathcal{U}$ $((\#_{top_{\tau_1}} \vee \#_{bot_{\tau_1}}) \wedge (\neg\#_{top_{\tau_2}} \wedge \neg\#_{bot_{\tau_2}}))$ |

The *expected reward* $\mathbb{E}(s_k, a_k)$ follows the classic Bellman formulation, combining immediate reward and future reward based on some discount factor $\gamma$ (see Appendix B for details). Intuitively, $\mathbb{E}(s_k, a_k)$ quantifies the long-term utility of taking action $a_k$ at state $s_k$. Finally, for any arbitrary state $s$ and action $a$, the Bellman Equation then defines the $Q$-value recursively for each $(s, a) \in S \times A$, denoted as $Q^{\mathcal{NN}}(s, a)$:

$$Q^{\mathcal{NN}}(s, a) \triangleq \sum_{s' \in S} \mathbf{P}(s, a, s') \left[ R(s, a) + \gamma \sum_{a' \in A} \mathcal{NN}(a' \mid s')\, \mathbb{E}(s', a') \right],$$

where $\mathcal{NN}(a' \mid s')$ indicates that $\mathcal{NN}$ takes action $a'$ on state $s'$. Consequently, the optimal action-value function for each $(s, a)$ is:

$$Q^{\mathcal{NN}^\star}(s, a) \triangleq \max_{\mathcal{NN}} Q^{\mathcal{NN}}(s, a) \tag{3}$$

Intuitively, $Q^{\mathcal{NN}^\star}(s, a)$ captures the maximum expected reward achievable from a state $s$ by taking action $a$. We remark that, our framework samples all paths for each quantifier simultaneously. That is, the learned neural network $\mathcal{NN}^\star$ induces a set of $n$ functions $\{\mathcal{NN}_1^\star, \dots, \mathcal{NN}_n^\star\}$, where $n = |Vars(\varphi)|$, and for all $1 \le \ell \le n$, $\mathcal{NN}_\ell^\star$ maps a state to an optimal actions for a path $\zeta_\ell$.

**Constructing Policies.** Based on the learned $\mathcal{NN}^\star$, we now construct the tuple of policies $\langle \pi_i^\star \rangle_{i \in \mathbb{Q}^\exists}$ and $\langle \pi_j^\star \rangle_{j \in \mathbb{Q}^\forall}$ that solves (2) for each $k$-step ranging over the sample size as follows.

- For each $j \in \mathbb{Q}^\forall$, we inductively construct the policies:

$$\pi_j^\star(\zeta_{j\,[0:k]}) \triangleq \mathcal{NN}_j^\star(s_k)$$

- For each $i \in \mathbb{Q}^\exists$, we construct a Skolem witness as follows:

$$\pi_i^\star(\zeta_{i\,[0:k]}) \triangleq \mathcal{NN}_i^\star \Big( \mathbf{f}_i \big( \mathsf{Tr}(\zeta_{i_1\,[0:k]}), \dots, \mathsf{Tr}(\zeta_{i_{|\mathbb{Q}_i^\forall|}\,[0:k]}) \big) \Big)$$

That is, the optimal policy for a finite prefix $\zeta_i$ with $i \in \mathbb{Q}^\exists$ depends on the optimal actions taken along the preceding universal paths $\zeta_{i_1}, \dots, \zeta_{i_{|\mathbb{Q}_i^\forall|}}$, illustrating how our framework captures the dependency of an existential path on the universally quantified ones. To this end, from the learned neural network $\mathcal{NN}^\star$, we succesfully derive two tuples $\langle \pi_i^\star \rangle_{i \in \mathbb{Q}^\exists}$ and $\langle \pi_j^\star \rangle_{j \in \mathbb{Q}^\forall}$ for Equation (2).

> **Theorem 2** *Given an MDP $\mathcal{M}$ and a HyperLTL formula $\varphi$, the optimal neural network function $\mathcal{NN}^\star$ derives a tuple of Skolem function witnesses $\langle \mathbf{f}_i \rangle_{i \in \mathbb{Q}^\exists}$ and a tuple of optimal policies $\langle \pi_j^\star \rangle_{j \in \mathbb{Q}^\forall}$ that optimize the satisfaction of $\mathbf{Skolem}(\varphi)$.*

Theorem 2 gives the premise of Theorem 1, which, in turn, solves the original problem stated in Section 3 (detailed proof in Appendix A).

## 6 Implementation and Experiments

HYPRL is fully implemented (see Figure 4). Given a HyperLTL formula $\varphi$ with $n$ quantifiers, HYPRL first constructs its Skolemized form $\mathbf{Skolem}(\varphi)$, as prescribed in Section 5.1. Next, at each step

**Table 2:** Descriptions of the case studies.

| | |
|---|---|
| Safe RL in Grid Worlds (SRL) | Two agents collaboratively learn policies $\langle \pi_1^\star, \pi_2^\star \rangle$ to reach their respective goals while avoiding collisions in a grid world. |
| Deep Sea Treasure (DST) | Two agents (a driver and a treasure collector) navigate among treasures. Learn policies $\langle \pi_1^\star, \pi_2^\star \rangle$ such that, for all ways the collector maximizes treasures, a safe route exists for the driver to exit within the time limit $\delta$. |
| Post Correspondence Problem (PCP) | Learn policies $\langle \pi_1^\star, \pi_2^\star \rangle$ for PCP: for any semi-matching sequence of dominos, there exists an extension into full matches. |

$t$, for each $i \in \{1, \ldots, n\}$, a policy $\pi_i$ observes a finite path $\zeta_{i[0:t]}$, and selects an action $\pi_i(\zeta_{i[0:t]})$, yielding an updated path $\zeta_{i[0:t+1]}$. The environment computes feedback using robustness function $\rho$ to guide learning (Section 5.2). The optimal tuple of policies $\langle \pi_1^\star, \ldots, \pi_n^\star \rangle$ are learned iteratively using a selected RL algorithm such as DQN [36], PPO [39], or CQ-Learning [15] (Section 5.3). We summarize all our case studies and their corresponding HyperLTL specifications in Table 2 and Table 1, respectively. We also investigate the wildfire scenario introduced in Section 1 with larger grid-worlds. All experiments are ran on an Apple M1 Max (10-core CPU, 24-core GPU). Additional details of experimental setups are provided in Appendix D.

**Baseline.** We experiment all cases using the setup from [32] by comparing against manually designed reward functions. To ensure a fair comparison, for DST, we use the reward function from [42]; for the rest of the cases, we construct several reward functions and report the best-performing one (details are provided in Appendix D). For SRL, we also compare with the method called "shield synthesis" [17], where a pre-synthesized *shield* enforces a safety property during learning and, in this specific case, can reason about inter-agent dependencies.

**Results and Analysis.** First, for SRL, we use the maps from [35] and compare CQ-learning+HYPRL against shield synthesis [17] (detailed map descriptions are inAppendix D). It is worth noting that for [17], a shield has to be explicitly designed to avoid unsafe states, whereas HYPRL achieves better safe and goal-directed behavior purely through the robustness values derived from a HyperLTL formula. Our analysis for SRL is presented in Table 3. In a two-agent setting, the result shows that the policies learned by HYPRL outperform the shielded agent in all of the environments by requiring fewer steps to reach the goal, except in SUNY, where their performance is quite similar. Furthermore, in both SUNY and MIT maps, HYPRL successfully avoids collisions as what we replicated from [17]. In a three-agent setting (where the HyperLTL formula is expanded to $\forall\forall\exists$), the shielding method successfully avoids all collisions; however, it failed to reach the goals within the bound of 100 steps in ISR, Pentagon, and MIT. In contrast, HYPRL was able to guide the agents to reach their goals significantly faster than both baselines and achieved fewer collisions compared to CQ-learning. We emphasize that although shield synthesis can avoid collision in all cases, it comes with the cost of spending a significant number of steps to reach the goal. In contrast, HYPRL maintains low collision rates in ISR and Pentagon, and learns policies that reach the goal with substantially fewer steps in ISR, Pentagon, and MIT maps.

Second, for SRL, we compare DQN+HYPRL with DQN and present the results in Figure 6. The top four figures demonstrate that HYPRL always achieves a higher number of successful goal completions under the same number of learning episodes. The bottom four figures show that HYPRL requires fewer episodes to learn collision avoidance.

For DST, we compare PPO+HYPRL with PPO, and DQN+HYPRL with DQN, using the reward functions from [42] in both cases. We evaluate using two metrics: (1) the total amount of collected treasures ($\sum$ 📦), and (2) the average amount of collected treasures per step ($\sum$ 📦 $/steps$). As shown

**Table 3:** SRL, avg. and std. error of 10 evaluations after 1k training episodes, 10 runs, where S/B stands for step-bound which is 100 steps for all cases.

| Maps | No. Agents | CQ | | CQ + Shield [17] | | CQ + HYPRL | |
|---|---|---|---|---|---|---|---|
| | | Steps | Collisions | Steps | Collisions | Steps | Collisions |
| ISR | | 27.95±7.4 | 0.19±0.1 | 17.40±2.2 | **0.00**±0.0 | **7.58**±0.3 | 0.25±0.2 |
| Pentagon | 2 | 36.46±7.7 | 0.28±0.1 | 75.20±12.6 | **0.00**±0.0 | **11.90**±4.6 | 0.53±0.5 |
| SUNY | | 11.99±0.5 | 0.01±0.0 | **11.50**±0.3 | **0.00**±0.0 | 12.48±0.6 | **0.00**±0.0 |
| MIT | | 41.28±8.5 | 0.20±0.1 | 33.46±3.4 | **0.00**±0.0 | **23.20**±0.5 | **0.00**±0.0 |
| ISR | | 98.79±0.8 | 12.68±3.8 | S/B | **0.00**±0.0 | **74.18**±5.1 | 7.78±1.0 |
| Pentagon | 3 | 97.15±2.4 | 16.46±7.2 | S/B | **0.00**±0.0 | **78.82**±1.7 | 10.92±1.4 |
| SUNY | | 84.89±7.9 | 0.63±0.2 | 82.35±4.1 | **0.00**±0.0 | **44.95**±8.3 | 0.71±0.4 |
| MIT | | 96.96±1.8 | 2.83±1.3 | S/B | **0.00**±0.0 | **71.53**±7.7 | 1.58±0.7 |

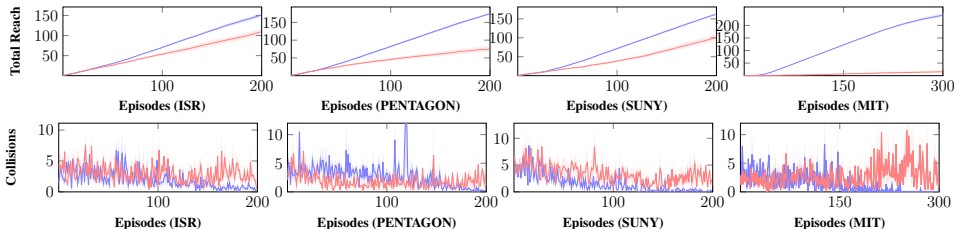

**Figure 6:** SRL results, avg. and std. error of total number of goal completions by 2 agents (top) and collisions (bottom). DQN+HYPRL (line ——) and DQN (line ——), over 10 runs.

**Table 4:** DST results, avg. and std. error over 10 evaluations across, 10 runs, and Epi. stands for "Episodes".

| Method | Epi. | $\sum$ | $\sum/steps$ | Method | Epi. | $\sum$ | $\sum/steps$ |
|---|---|---|---|---|---|---|---|
| PPO | 500 | 4.31±1.2 | 0.17±0.04 | DQN | 500 | 1.08±0.2 | 0.05±0.00 |
| PPO + HYPRL | 500 | **22.93**±2.2 | **0.91**±0.08 | DQN + HYPRL | 500 | **4.12**±1.4 | **0.14**±0.05 |
| PPO | 1000 | 3.22±0.8 | 0.12±0.03 | DQN | 1000 | 1.45±0.2 | 0.02±0.00 |
| PPO + HYPRL | 1000 | **23.97**±2.2 | **1.12**±0.08 | DQN + HYPRL | 1000 | **4.43**±0.8 | **0.21**±0.03 |

**Table 5:** Wildfire results, avg. and std. error of 10 trials after 5k episodes, 10 runs.

| Size | Method | Dist | Steps $O_1$ | Steps $O_2$ |
|---|---|---|---|---|
| $3^2$ | PPO | 2.5±0.01 | 33.43±4.1 | 787.03±31.8 |
| | PPO + HYPRL | **2.30**±0.03 | **18.940**±1.1 | **143.550**±1.1 |
| $5^2$ | PPO | 4.2±0.01 | 62.7±9.7 | S/B |
| | PPO + HYPRL | **2.1**±0.05 | **59.50**±14.3 | **8057.5**±121.4 |
| $8^2$ | PPO | 11.2±0.03 | 16801.8±2144.0 | S/B |
| | PPO + HYPRL | **6.94**±0.07 | **4149.6**±1743.1 | **386.2**±80.5 |
| $10^2$ | PPO | 10.9±0.01 | S/B | 29023.6±976.4 |
| | PPO + HYPRL | **5.3**±0.10 | **21272.8**±3579.0 | **570.3**±52.2 |

in Table 4, policies learned by HYPRL outperform those learned by the baseline, regardless of the number of training episodes or the choice of the RL algorithm.

Finding policies to solve the PCP problem is inherently challenging due to its undecidability. HYPRL is able to learn optimal policies that solve PCP more efficient, compare to the baseline. Figure 5 presents the performance comparison of DQN+HYPRL with DQN, showing that HYPRL achieves more successful matches in both 5- and 6-domino settings.

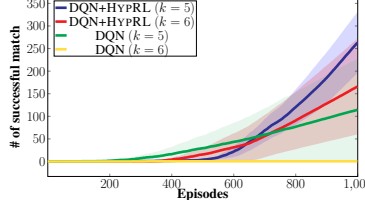

**Figure 5:** PCP results, avg. and std. error of total matches, 10 runs.

Lastly, we evaluate HYPRL on the wildfire scenario and compare PPO+HYPRL with PPO. As shown in Table 5, even when scaling up the grid-world environment from $3^2$ to $5^2$, $8^2$, and $10^2$, HYPRL consistently outperforms the baseline by requiring fewer steps to rescue victims, extinguish all fires, and maintain a safe distance between the two agents. The step-bounds are set to 10k steps for the $3^2$ and $5^2$ environments, 20k steps for $8^2$, and 30k steps for $10^2$. In fact, PPO fails to learn policies capable of rescuing victims ($O_2$) within the step-bounds for $5^2$ and $8^2$, and is unable to extinguish all fires ($O_1$) within the step-bound in the $10^2$ setting.

## 7 Conclusion

We introduced HYPRL, a reinforcement learning framework that synthesizes control policies for tasks specified as HyperLTL formulas over MDPs with unknown transitions. By leveraging the expressiveness of hyperproperties, HYPRL can handle complex multi-agent objectives and relational constraints, including safe planning and undecidable problems like PCP. Our quantitative semantics automatically derives robustness values from the specification and learns control policies that maximize the probability of satisfying the given HyperLTL property. Implemented with off-the-shelf RL algorithms, HYPRL demonstrates superior performance over comparable baselines across a set of diverse case studies, especially in scenarios where reward shaping is non-trivial.

**Limitations.** HYPRL currently does not support fully decentralized control policies, where each agent learns its local policy without access to global observations and rewards. Second, it is possible that environment states are not fully observable. That is, instead of an MDP, the environment is a partially observable MDP (POMDP). We believe these limitations open up exciting future work and can be addressed by extending HYPRL.

**Societal impacts.** This work marks a significant step towards extending MARL to complex tasks with relational requirements. By enabling the learning of policies under interdependent objectives, our approach broadens the applicability of MARL to critical domains such as disaster response, swarm drones, smart home automation, and industrial resource management, where coordination among agents is essential for success.

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

# A  Proofs of Theorems

## A.1  Proof of Theorem 1

Recall that given a HyperLTL formula $\varphi$, $\textbf{Skolem}(\varphi)$ is in the form of:

$$\textbf{Skolem}(\varphi) = \underbrace{\exists \mathbf{f}_i(\tau_{i_1}, \ldots, \tau_{i_{|\mathbb{Q}_i^\forall|}})}_{\text{for each } i \in \mathbb{Q}^\exists} . \quad \underbrace{\forall \tau_j.}_{\text{for each } j \in \mathbb{Q}^\forall} \quad \textbf{Skolem}(\psi)$$

Assuming that $\langle \pi_i^\star \rangle_{i \in \mathbb{Q}^\exists} \cup_{\le} \langle \pi_j^\star \rangle_{j \in \mathbb{Q}^\forall}$ is the optimal set of policies for the Skolemized $\varphi$, then for every $k \ge 0$, it maximizes the probability of converging to $\rho_{max}$ for the inner LTL sub-formula $\textbf{Skolem}(\psi)$:

$$\mathbb{P}\Big[\rho\Big(\text{zip}\big(\langle \text{Tr}(\zeta_{i\,[0:k_i]})\rangle\big) \cup_{\le} \langle \text{Tr}(\zeta_{j\,[0:k_j]})\rangle\big), \textbf{Skolem}(\psi)\Big)\Big]_{i \in \mathbb{Q}^\exists, j \in \mathbb{Q}^\forall} \xrightarrow{\star} \rho_{max}$$

That is, on any step $k$, the zipped path derives a set of paths that instantiate each path variables in the original $\varphi$ as follows:

- For all universal quantified paths $\tau_j$, where $j \in \mathbb{Q}^\forall$:

$$[\tau_j \mapsto \text{Tr}(\zeta_{j\,[0:k]})]$$

- For all existential quantified paths $\tau_i$, where $i \in \mathbb{Q}^\exists$:

$$[\tau_i \mapsto \mathbf{f}_i\big(\text{Tr}(\zeta_{i_1\,[0:k]}), \ldots, \text{Tr}(\zeta_{i_{|\mathbb{Q}_i^\forall|}\,[0:k]})\big)], \text{ and}$$

$$\text{for each } \ell \in \{1, \ldots, |\mathbb{Q}_i^\forall|\}, \text{ if } i_\ell = j \text{ then } \text{Tr}(\zeta_{i_\ell\,[0:k]}) = \text{Tr}(\zeta_{j\,[0:k]})$$

This instaintiation guarantees that the probability of satisfying $\varphi$ is maximized up to step $k$. Next, the zipped path proceeds to step $k+1$ with an optimal action given by its corresponding optimal policy $\langle \pi_i^\star \rangle_{i \in \mathbb{Q}^\exists} \cup_{\le} \langle \pi_j^\star \rangle_{j \in \mathbb{Q}^\forall}$. That is, the same path mappings of each universally quantified path variable $\zeta_j$ and existentially quantified Skolem function witnesses $\mathbf{f}_i$ holds for all $k \ge 0$. Finally, for the original HyperLTL formula $\varphi = \mathbb{Q}_1\tau_1.\mathbb{Q}_2\tau_2.\ldots.\mathbb{Q}_n.\tau_n.\ \psi$, by instantiating the trace variables in the same fashion for all $\tau_i$ of $i \in \mathbb{Q}^\exists$ and all $\tau_j$ of $j \in \mathbb{Q}^\forall$, the same optimality immediately follows, because the optimal convergence to $\rho_{max}$ means maximizing the probability of satisfying the inner LTL sub-formula. That is, the optimal set of paths (derived from the zipped path) also optimizes the satisfaction of original $\varphi$ for all steps $k \ge 0$. To this end, we proved that, an optimal tuple of policies $\langle \pi_i^\star \rangle_{i \in \mathbb{Q}^\exists} \cup_{\le} \langle \pi_j^\star \rangle_{j \in \mathbb{Q}^\forall}$ for $\textbf{Skolem}(\varphi)$ is also an optimal set of policies for $\varphi$, that optimizes the probability of satisfying $\varphi$ in $\mathcal{M}$.

## A.2  Proof of Theorem 2

Bellman's Principle of Optimality [6] states that "*for an optimal policy, no matter what the initial decision is, the remaining decisions must constitute an optimal policy with regard to the state resulting from the initial decision*". An extended lemma (proved in [1]) states that for a discounted MDP, there exists an *optimal* policy, denoted as $\mathcal{NN}^\star$, such that for all $(s,a) \in S \times A$, there exists a maximum Q-value achieved by $\mathcal{NN}^\star$ (denoted as $Q^{\mathcal{NN}^\star}$) as introduced in Equation (3):

$$Q^{\mathcal{NN}^\star}(s,a) \triangleq \max_{\mathcal{NN}} Q^{\mathcal{NN}}(s,a)$$

Let us denote $\mathcal{NN}(a \mid s)$ as $\mathcal{NN}$'s decision to take action $a$ on state $s$. Our goal is to find an optimized $\mathcal{NN}$, such that:

$$\max_{\mathcal{NN}} \sum_{s' \in S} \mathbf{P}(s,a,s')\Big[R(s,a) + \gamma \sum_{a \in A} \mathcal{NN}(a \mid s)\mathbb{E}(s',a')\Big]$$

where $\mathbf{P}(s,a,s')$ is the one-step transition probability, $R(s,a)$ is the reward of taking action $a$ on state $s$, $\gamma$ is a discount selected factor, and $\mathbb{E}$ is the expected reward of keep taking an action $a'$ on a state $s'$. (as defined in Section 5.3). Recall that in Section 5.2, our immediate reward $R(s,a)$ is associated with a robustness value of a finite prefix by evaluation only up to the current seen state

$s$ (i.e., independent from the unseen $s'$ after taking action $a$). As a result, maximizing $\mathcal{N}\mathcal{N}$ do not depend on $R(s, a)$, so the previous optimization problem is equivalent to:

$$R(s, a) + \max_{\mathcal{N}\mathcal{N}} \left[ \sum_{s' \in S} \mathbf{P}(s, a, s') \, \gamma \sum_{a \in A} \mathcal{N}\mathcal{N}(a \mid s) \mathbb{E}(s', a') \right]$$

Let us now only focus on the optimization part (i.e., the right side of the plus operator in the above formula). Based on the definition of expected reward defined in Section 5.3, the above formula shows that an optimal $\mathcal{N}\mathcal{N}^\star$ is more likely to the take action $a$ (by the learned $\mathcal{N}\mathcal{N}(a \mid s)$) that has higher probability (decided by $\mathbf{P}(s, a, s')$) to transit to an unseen state $s'$ that lead to higher expected value (estimated by $\mathbb{E}(s', a')$). That is, given a state $s$, $\mathcal{N}\mathcal{N}^\star$ outputs an optimal action $a$, such that:

$$\gamma \mathbb{E}_{s' \sim \mathbf{P}(s, a, \cdot))} \big[ R(s', a') \big]^{\mathcal{N}\mathcal{N}^\star},$$

which, intuitively, represents an optimal *one-step* look-ahead. Now, to connect the above formula with the reward function defined in Section 5.3, we have:

$$\gamma \mathbb{E}_{s' \sim \mathbf{P}(s, a, \cdot))} \left[ \rho \Big( \mathsf{zip}\big( \langle \mathsf{Tr}(\zeta_{i_{[0:k]}}) \rangle \rangle \cup_{\leq} \langle \mathsf{Tr}(\zeta_{j_{[0:k]}}) \rangle \big), \mathbf{Skolem}(\psi) \Big) \right]^{\mathcal{N}\mathcal{N}^\star}_{i \in \mathbb{Q}^{\exists}, j \in \mathbb{Q}^{\forall}},$$

where $s$ is the state on the $k$-th step of the zipped path. Recall that $\rho$ is constructed using min-max approach as presented in Figure 3, so the optimal outcome of $\rho$ can be derived from the *Minimax Lemma* [18]. That is, if each path always considers the "worst-possible" scenario that other paths will act during learning, it leads to a set of optimal policy among all paths. Hence, $\rho$ is guaranteed optimal for all steps $k \geq 0$, which implies the maximum probability of the following:

$$\mathbb{P} \left[ \rho \Big( \mathsf{zip}\big( \langle \mathsf{Tr}(\zeta_{i_{[0:k_i]}}) \sim \mathcal{D}_{\pi_i} \rangle \rangle \cup_{\leq} \langle \mathsf{Tr}(\zeta_{j_{[0:k_j]}}) \sim \mathcal{D}_{\pi_j} \rangle \big), \mathbf{Skolem}(\psi) \Big) \xrightarrow{\star} \rho_{max} \right]_{i \in \mathbb{Q}^{\exists}, j \in \mathbb{Q}^{\forall}}$$

To this end, we prove that the action chose by $\mathcal{N}\mathcal{N}^\star$ achieves the maximum expected value $\mathbb{E}$ for all $(s, a) \in S \times A$. Finally, $\langle \mathbf{f}_i \rangle_{i \in \mathbb{Q}^{\exists}}$ and $\langle \pi_j^\star \rangle_{j \in \mathbb{Q}^{\forall}}$ can be inductively constructed from $\mathcal{N}\mathcal{N}^\star$ (as we elaborated in Section 5.3), which is a policies set that optimizes the satisfaction of $\mathbf{Skolem}(\varphi)$.

## B  Additional Algorithmic Details of HypRL

**Expected Reward of a Zipped Trace.**  The *expected reward* of a state $s_k$ after taking an action $a_k$ with *discount factor* $\gamma \in [0, 1]$ is:

$$\mathbb{E}(s_k, a_k) \triangleq \sum_{t=0}^{\infty} \gamma^t \times R(s_{k+t+1}, a_{k+t+1})$$

Intuitively, $\mathbb{E}(s_k, a_k)$ evaluates "how good" of choosing $a_k$ on $s_k$ in infinite time steps, where each step is *discounted* by $\gamma$. We address that, $\gamma$ is often chosen based on the optimization goal. For example, for short-term tasks, a lower $\gamma$ is preferred because the expected robustness value focuses more on immediate $\rho$ value. We report the setup of these hyperparameters in Appendix D.

## C  A Comprehensive Example of HypRL Algorithm

In this section, we present a comprehensive head-to-toe example that aligns with the algorithmic details presented in Section 5.

**Formula Skolemization.**  In Section 3, we presented the HyperLTL formula for our wildfire example as follows:

$$\textit{Requirements}: \ \varphi_{\mathsf{Rescue}} \triangleq \forall \tau_1. \exists \tau_2. (\psi_{\mathsf{fire}} \wedge \psi_{\mathsf{save}} \wedge \psi_{\mathsf{dist}} \wedge \psi_{\mathsf{safe}})$$

$$O_1: \ \psi_{\mathsf{fire}} \triangleq \Diamond(i_{\tau_1}) \wedge \Diamond(f_{\tau_1}) \wedge \Diamond(c_{\tau_1}) \qquad C_1: \ \psi_{\mathsf{dist}} \triangleq \Box(|\mathsf{Location}_{\tau_1} - \mathsf{Location}_{\tau_2}| < 3)$$

$$O_2: \ \psi_{\mathsf{save}} \triangleq \Diamond(g_{\tau_2}) \wedge \Diamond(f_{\tau_2}) \qquad\qquad\quad C_2: \ \psi_{\mathsf{safe}} \triangleq (\neg i_{\tau_2} \, \mathcal{U} \, i_{\tau_1}) \wedge (\neg f_{\tau_2} \, \mathcal{U} \, f_{\tau_1}) \wedge (\neg c_{\tau_2} \, \mathcal{U} \, c_{\tau_1})$$

Following the illustration in Section 5.1, the Skolemized form of the formula $\varphi_{\text{Rescue}}$ is as follows:

$$\textbf{Skolem}(\varphi_{\text{Rescue}}) \triangleq \exists \mathbf{f}_2(\tau_1).\forall \tau_1.$$

$$\textbf{Skolem}(\psi_{\text{fire}}) \wedge \textbf{Skolem}(\psi_{\text{save}}) \wedge \textbf{Skolem}(\psi_{\text{dist}}) \wedge \textbf{Skolem}(\psi_{\text{safe}})$$

$$\textbf{Skolem}(\psi_{\text{fire}}) \triangleq \Diamond(i_{\tau_1}) \wedge \Diamond(f_{\tau_1}) \wedge \Diamond(c_{\tau_1})$$

$$\textbf{Skolem}(\psi_{\text{save}}) \triangleq \Diamond(g_{\mathbf{f}_2}) \wedge \Diamond(f_{\mathbf{f}_2})$$

$$\textbf{Skolem}(\psi_{\text{dist}}) \triangleq \Box(|\text{Location}_{\tau_1} - \text{Location}_{\mathbf{f}_2}| < 3)$$

$$\textbf{Skolem}(\psi_{\text{safe}}) \triangleq (\neg i_{\mathbf{f}_2} \, \mathcal{U} \, i_{\tau_1}) \wedge (\neg f_{\mathbf{f}_2} \, \mathcal{U} \, f_{\tau_1})(\neg c_{\mathbf{f}_2} \, \mathcal{U} \, c_{\tau_1})$$

In this context, there are two quantifiers, $\mathbb{Q}_2 = \exists$ so $\mathbb{Q}^\exists = \{2\}$, and $\mathbb{Q}_1 = \forall$ so $\mathbb{Q}^\forall = \{1\}$. Besides, all propositions of the existential quantified path variable (originally subscripted by $\tau_2$) average now subscripted by $\mathbf{f}_2$ in the Skolemized form. Furthermore, for each inner LTL sub-formula, the propositions subscripted with $\tau_2$ is substituted with propositions subscripted with $\mathbf{f}_2$.

**Robustness Optimization.** Continuing with the Skolemized formula:

$$\exists \mathbf{f}_2(\tau_1).\forall \tau_1. \, \textbf{Skolem}(\psi_{\text{fire}}) \wedge \textbf{Skolem}(\psi_{\text{save}}) \wedge \textbf{Skolem}(\psi_{\text{dist}}) \wedge \textbf{Skolem}(\psi_{\text{safe}})$$

Our goal is to optimize the following:

$$\langle \pi_1^\star, \pi_2^\star \rangle \in \underset{\langle \pi_1, \pi_2 \rangle}{\arg\max} \, \mathbb{P}\Big[\rho\big(\textsf{zip}(\langle \textsf{Tr}(\zeta_{1[0:k_1]} \sim \mathcal{D}_{\pi_1}), \textsf{Tr}(\zeta_{2[0:k_2]} \sim \mathcal{D}_{\pi_2})\rangle),$$

$$\textbf{Skolem}(\psi_{\text{fire}} \wedge \psi_{\text{save}} \wedge \psi_{\text{dist}} \wedge \psi_{\text{safe}})\big) \overset{\star}{\to} \rho_{max}\Big]$$

Notice that, by applying $\cup_\leq$, the order of the quantified traces are in the right order with respect to the original HyperLTL formula $\varphi$.

# D Implementation and Experimental Details

## D.1 From HyperLTL Formulas to Robustness Functions

In our implementation, we use the theory developed in Section 5.1 to obtain the Skolemized form of the formula, denoted $\textbf{Skolem}(\varphi)$. We then generate the corresponding robustness function using the approach presented in Section 5.2 (see [32] for details on translating temporal properties into robustness functions). We now elaborate the following term from the quantitative semantics introduced in Figure 3:

$$\rho\big(\textsf{Tr}(\zeta_{[\ell:k]}), f\big(L(s_\ell) < c\big)\big) = c - f\big(L(s_\ell)\big)$$

This equation lies at the core of transitioning from logical properties (which yield Boolean outputs) to robustness functions (which produce real-valued outputs). To apply this transformation, we must define the function $f$ and the constant $c$ for each property considered in this paper. We now provide details explanations on how we define such functions and constants for each formula introduced in this paper.

- The HyperLTL formula defined for safe reinforcement learning (SRL), denoted as $\varphi_{\text{SRL}}$:
  - For the property $\langle x_{\tau_1}, y_{\tau_1} \rangle = \langle x_{G1}, y_{G1} \rangle$, we define $f$ as a distance function and set $c = 1$. The distance function we have in this case study follows the well-known Manhattan distance definition. That is:

    $$\texttt{Dist}(\langle x_{\tau_1}, y_{\tau_1} \rangle, \langle x_{G1}, y_{G1} \rangle) \triangleq (|x_{\tau_1} - x_{G1}| + |y_{\tau_1} - y_{G1}|)$$

    The, by setting $c = 1$, this yields the following expression:

    $$\texttt{Dist}(\langle x_{\tau_1}, y_{\tau_1} \rangle, \langle x_{G1}, y_{G1} \rangle) < 1,$$

    which expresses that the distance from the first agent (i.e., $\tau_1$) to its goal (i.e., $G_1$) is less than one, which implies that the first agent has *indeed* reached the specified goal location. The corresponding equation is:

    $$1 - \texttt{Dist}(\langle x_{\tau_1}, y_{\tau_1} \rangle, \langle x_{G1}, y_{G1} \rangle)$$

    The optimal value is achieved when the agent is exactly at the location of the goal position, giving $1 - 0 = 1$.

- For the property $\langle x_{\tau_2}, y_{\tau_2} \rangle = \langle x_{G2}, y_{G2} \rangle$, the same definition for the distance function is similarly defined. That is:

$$\texttt{Dist}(\langle x_{\tau_2}, y_{\tau_2} \rangle, \langle x_{G2}, y_{G2} \rangle) < 1,$$

  which expresses that the distance from the second agent to its goal is less than one, so the corresponding equation is:

$$1 - \texttt{Dist}(\langle x_{\tau_2}, y_{\tau_2} \rangle, \langle x_{G2}, y_{G2} \rangle)$$

  The optimal value is achieved when the second agent is exactly at the goal position, giving $1 - 0 = 1$.

- For the property $\langle x_{\tau_1}, y_{\tau_1} \rangle \neq \langle x_{\tau_2}, y_{\tau_2} \rangle$, we define $f$ as a distance function and set $c = -1$. This yields the condition:

$$-\texttt{Dist}(\langle x_{\tau_1}, y_{\tau_1} \rangle, \langle x_{\tau_2}, y_{\tau_2} \rangle) < -1,$$

  which expresses that the distance between the first and second agent (i.e., $\tau_1$ and $\tau_2$) is greater than one; i.e., the agents are not in the same location. The corresponding equation is:

$$-1 + \texttt{Dist}(\langle x_{\tau_1}, y_{\tau_1} \rangle, \langle x_{\tau_2}, y_{\tau_2} \rangle).$$

  This equation is maximized when the two agents are far apart, and it evaluates to $-1$ when they are in same location.

- The HyperLTL formula defined for deep sea treasure (DST), denoted as $\varphi_{\text{DST}}$:

  - For the properties *T1*, *T2*, $\cdots$, we define $f$ for each treasure *Ti* a distance function between the position of the submarine driver ($\tau_i$) and the treasure location (*Ti*), and set $c = 1$. This yields the condition:

$$\texttt{Dist}(\langle x_{\tau_1}, y_{\tau_1} \rangle, \langle x_{T1}, y_{T1} \rangle) < 1$$

    which expresses that the distance from the first agent to *T1* is less than one. The corresponding equation is:

$$1 - \texttt{Dist}(\langle x_{\tau_1}, y_{\tau_1} \rangle, \langle x_{T1}, y_{T1} \rangle)$$

    The same approach is applied to *T2*, *T3*,... and the remaining properties.

  - For the property $step_{\tau_2} < \delta$, we use the same approach and obtain the following equation:

$$\delta - step_{\tau_2}$$

  - For the property $|pos_{\tau_1} - pos_{\tau_2}| < 1$, we define $f$ as a distance function and set $c = 1$. This yields the condition:

$$\texttt{Dist}(\langle x_{\tau_1}, y_{\tau_1} \rangle, \langle x_{\tau_2}, y_{\tau_2} \rangle) < 1,$$

    which expresses that the distance between the first and second agent is less than one. That is, the agents are in the same location. The corresponding equation is:

$$1 - \texttt{Dist}(\langle x_{\tau_1}, y_{\tau_1} \rangle, \langle x_{\tau_2}, y_{\tau_2} \rangle)$$

    This equation is maximized when the agents are in the same location.

- The formula defined for the Post Correspondence Problem (PCP), denoted as $\varphi_{\text{PCP}}$:

  - To address the properties such as $(p_1 \leftrightarrow p_2)$, where $p$ means a sequence of alphabets (that forms a string), we define $f$ as a function of evaluating whether two strings are matching and set $c = 1$. The matching function we have in this case study is as follows:

$$\texttt{Match}(p_1, p_2) = \begin{cases} 0 & \text{if } p_1 \text{ and } p_2 \text{ are equal,} \\ 1 & \text{otherwise.} \end{cases}$$

    This yields the condition:

$$\texttt{Match}(p_1, p_2) < 1,$$

    which expresses whether the two compared alphabets $p_1$ and $p_2$ are equal or not. The corresponding equation is:

$$1 - \texttt{Match}(p_1, p_2)$$

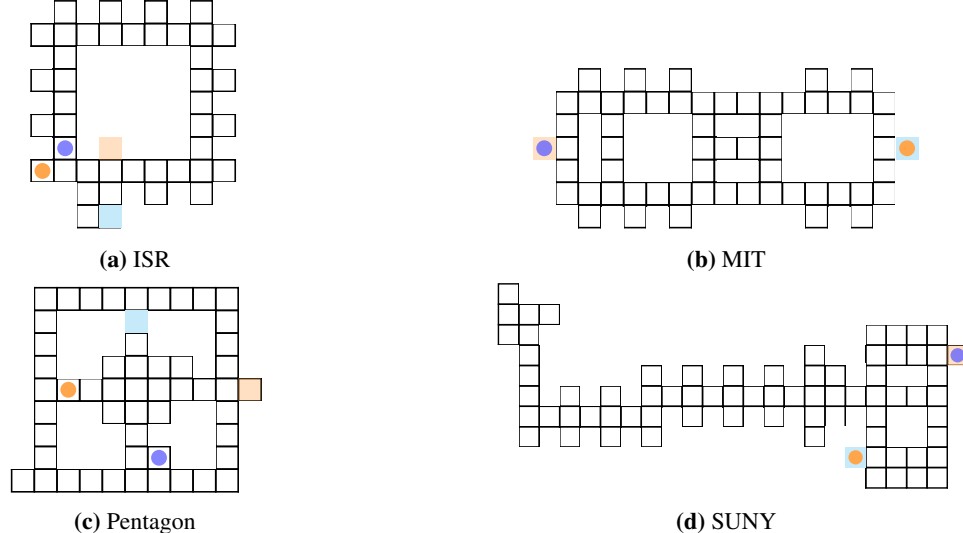

**(a)** ISR

**(b)** MIT

**(c)** Pentagon

**(d)** SUNY

**Figure 7:** Maps of Grid World [35] benchmarks.

- – For the symbol #, we are checking whether the alphabet under consideration is equal to #. This is expressed as the logical condition $p \leftrightarrow \#$.

- The HyperLTL formula defined for the wildfire scenario, denoted as $\varphi_{\mathsf{Rescue}}$:
  - – For the properties *V1*, *V2*, $\cdots$ and *F1*, *F2*, $\cdots$, we use the same distance function Dist (as in SRL and DST) to calculate the distance from agents to victims and fire zones, with $c = 1$. That is, the reachability of the two objectives.
  - – For the property $|\mathsf{Location}_{\tau_1} - \mathsf{Location}_{\tau_2}| < 3$, we use the distance function Dist with $c = 3$. This constraints make sure that for the optimized strategy, the two agents are always staying within the safe communication range.

### D.2 Safe RL (SRL)

**Case Study Setup.** In the Safe RL case study (see Figure 7), the blue circle (denoted A1) and the orange circle (denoted A2) are agents that aim to learn optimal policies to navigate from their initial positions to their respective targets: the blue square (G1) and the orange square (G2), while avoiding collisions. The state space is represented as a tuple $\langle x, y \rangle$, and the action space is defined as $a = \{stay, up, down, left, right\}$. The following HyperLTL formula $\varphi_{\mathsf{SRL}}$ specifies the required objectives:

$$\varphi_{\mathsf{SRL}} \triangleq \forall \tau_1. \exists \tau_2. \left( \Diamond \psi_{\mathsf{G1}_{\tau_1}} \wedge \Diamond \psi_{\mathsf{G2}_{\tau_2}} \wedge \Box \psi_{\mathsf{CA}_{\tau_1, \tau_2}} \right)$$

where the subformulas are defined as:

$$\psi_{\mathsf{G1}_{\tau_1}} \triangleq \langle x_{\tau_1}, y_{\tau_1} \rangle = \langle x_{G1}, y_{G1} \rangle, \qquad \psi_{\mathsf{G2}_{\tau_2}} \triangleq \langle x_{\tau_2}, y_{\tau_2} \rangle = \langle x_{G2}, y_{G2} \rangle,$$

$$\psi_{\mathsf{CA}_{\tau_1, \tau_2}} \triangleq \langle x_{\tau_1}, y_{\tau_1} \rangle \neq \langle x_{\tau_2}, y_{\tau_2} \rangle.$$

Here, $\psi_{\mathsf{G1}_{\tau_1}}$ and $\psi_{\mathsf{G2}_{\tau_2}}$ express that A1 and A2 eventually reach G1 and G2, respectively. The subformula $\psi_{\mathsf{CA}_{\tau_1, \tau_2}}$ ensures that the agents avoid collisions while navigating toward their goals.

**First Experminemt Setup.** In the first experiment, to ensure a fair comparison, we evaluate HYPRL combined with CQ-Learning against the shielding method [17] also using CQ-Learning. The implementation and hyperparameters of CQ-Learning are taken from [17]. Since CQ-Learning initially employs a decentralized phase (i.e., a single-agent state space), we modify the formula $\varphi_{\mathsf{SRL}}$ to remove inter-agent dependencies:

$$\forall \tau_1. \forall \tau_2. \left( \Diamond \psi_{\mathsf{G1}_{\tau_1}} \wedge \Diamond \psi_{\mathsf{G2}_{\tau_2}} \wedge \Box \psi_{\mathsf{CA}_{\tau_1, \tau_2}} \right)$$

In this setting, Skolemization is unnecessary, and the agents act independently. When the algorithm transitions into a joint phase, expanding the state space to include inter-agent interactions, we use the original specification $\varphi_{\text{SRL}}$. At this stage, HYPRL guides the learning process effectively to achieve both goal completion and collision avoidance.

**Second Experminemt Setup.** We employ DQN as our learning algorithm, utilizing a neural network with three hidden layers of 512 nodes and ReLU activation functions. We set the discount factor to $\gamma = 1.0$, the learning rate to 0.001, the initial $\epsilon$ to 1.0 with a decay rate of 0.995 down to a minimum of 0.01, and use the Adam optimizer. We set the number of training episodes to 200 for the SUNY, ISR, and Pentagon maps, and to 300 for the MIT map due to its increased complexity. Each episode consists of 300 steps (see Figure 7 for the map setups).

**Baselines Reward Functions.** The manually designed reward functions have the following form:

$$R^{\text{SRL}} = \begin{cases} a & \text{if both agents reach their respective goals,} \\ b & \text{if one agent reaches its goal,} \\ c & \text{if the agents collide.} \end{cases}$$

The function $R^{\text{SRL}}$ addresses all objectives and safety constraints of the problem. The manually designed reward function is tested with the following values:

- $a = 10, b = 5, c = -5$ ($R_1^{\text{SRL}}$)
- $a = 2, b = 1, c = -1$ ($R_2^{\text{SRL}}$)
- $a = 20, b = 10, c = -10$ ($R_3^{\text{SRL}}$)
- $a = 100, b = 50, c = -10$ ($R_4^{\text{SRL}}$)
- $a = 10, b = 5, c = -10$ ($R_5^{\text{SRL}}$)

In the main paper, we report the results obtained using values $a = 10, b = 5, c = -5$ ($R_1^{\text{SRL}}$) for the baseline reward function. Figure 8 presents the results corresponding to all different reward functions $R^1$, $R^2$, $R^3$, $R^4$, and $R^5$ we introduced above.

### D.3 Deep Sea Treasure (DST)

**Case Study Setup.** This case study (see Figure 9) was originally proposed by [42]. We adopt the implementation provided in [46]. The state space is represented as a tuple $\langle x, y, step \rangle$, and the action space is defined as $a = \{up, down, left, right\}$. The following HyperLTL formula $\varphi_{\text{DST}}$ specifies the required objectives:

$$\varphi_{\text{DST}} \triangleq \forall \tau_1. \exists \tau_2. \Diamond (T1_{\tau_1} \wedge \Diamond (T2_{\tau_1} \wedge \Diamond (T3_{\tau_1}) \dots)) \wedge \Box (step_{\tau_2} < \delta) \wedge \Box (|pos_{\tau_1} - pos_{\tau_2}| < 1)$$

The original case study is in a single-agent environment with two objectives. We modified it into a multi-agent problem by assigning one agent with the objective of maximizing collected treasure, and another agent with the task of keeping the number of steps below a certain threshold.

**Experminemt Setup.** For our experiments, we used PPO and DQN implementations from [38]. For DQN, we set the hyperparameters as follows: discount factor $\gamma = 0.99$, $\epsilon$ decaying from 1.0 to 0.07 over a fraction of 0.2, and a learning rate of 0.0004. Since the DQN implementation from [38] does not directly support multi-discrete actions, we expanded the action spaces of both agents and defined a unified action space compatible with this implementation. For PPO, we set hyperparameters as following, $\gamma = 0.99$, clipping factor to 0.2, learning rate to 0.0003, and GAE lambda to 0.98. Additionally, to ensure a fair comparison, we added a reward penalty of $-10$ to the baseline reward function whenever the two agents became separated.

### D.4 Additional Experiment on DST

We conduct an additional experiment on DST, where the focus is on minimizing the step constraint in the problem. In Table 6, we report the number of steps required for the cumulative treasure achieved by the treasure collector to exceed values of 1, 5, 10, 15, 20, and 30. The results show that HYPRL consistently and significantly outperforms the baselines in all settings, except when using PPO trained for 1000 episodes.

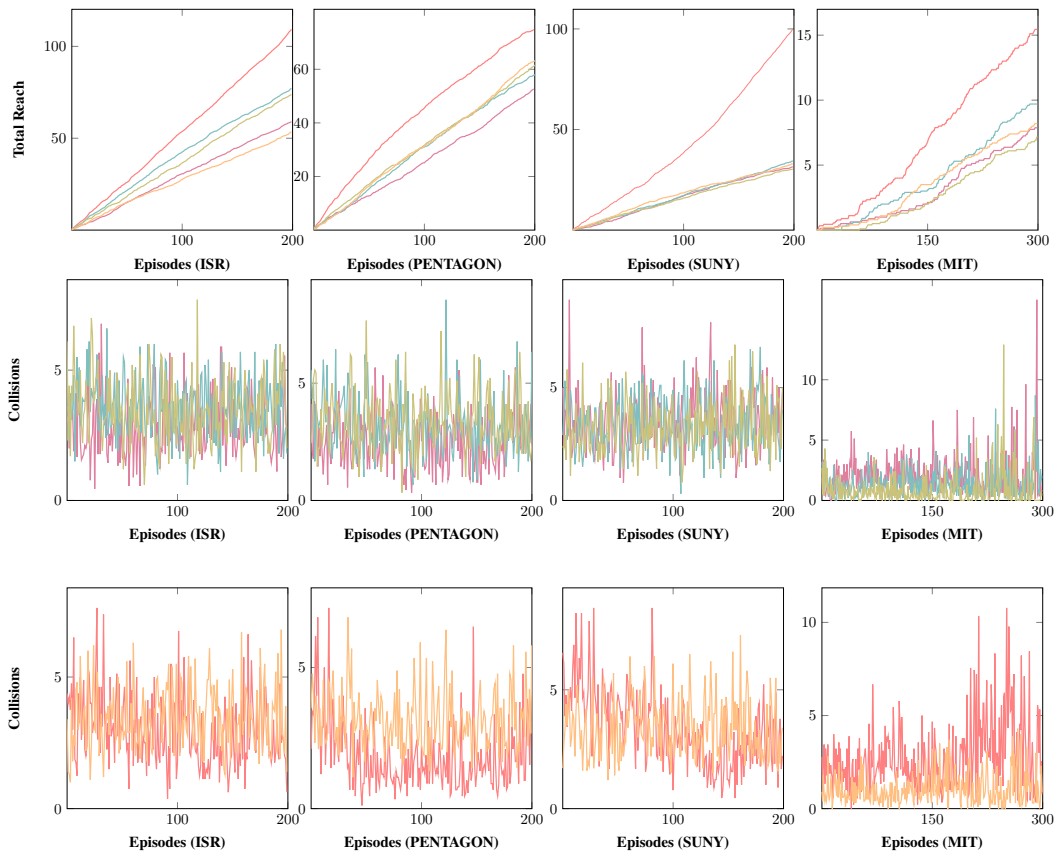

**Figure 8:** Total number of successful goal completions by both agents (top) and number of collisions (bottom). Lines ——, ——, ——, ——, and —— correspond to reward functions $R^1$, $R^2$, $R^3$, $R^4$, and $R^5$, respectively. Results are averaged over 10 runs.

### D.4.1 The Post Correspondence Problem (PCP)

**Case Study Setup.** PCP (see Figure 11) consists of a set of $k$ dominos, denoted as $D = \{dom_0, dom_1, \ldots, dom_k\}$. Each domino $dom_i$ ($0 \le i \le k$) is represented by a pair of nonempty finite words $(top_i, bot_i)$ from a given alphabet $\Sigma$. The objective is to find a finite sequence of dominos such that the concatenated words on the top match those on the bottom. In our case study, the state space is defined as a tuple $\langle top, bot \rangle$ for each domino. The action space consists of selecting a domino from $D$, resulting in $k$ possible actions $A = \{dom_0, dom_1, \ldots, dom_k\}$. Notably, traces (i.e., agents) are unaware of the context of the dominos and can only identify them by their labels (i.e., $dom_i$). The initial state of the MDP encodes empty words on both the top and bottom: $s^0 = \langle \varepsilon, \varepsilon \rangle$.

**Table 6:** DST results on steps required to achieve the desired treasures, showing average and standard error over 10 evaluations.

| Method | Episodes | Treasures Achieved | | | | | |
|---|---|---|---|---|---|---|---|
| | | 1 | 5 | 10 | 15 | 20 | 30 |
| PPO | 500 | 199.01±6.65 | 218.66±6.67 | 244.87±6.68 | 247.89±6.67 | 253.16±6.66 | 268.01±6.70 |
| PPO +HYPRL | 500 | **13.30**±1.01 | **29.72**±1.57 | **39.10**±1.81 | **49.22**±1.96 | **56.59**±2.03 | **71.55**±2.26 |
| PPO | 1000 | **5.75**±0.30 | **6.75**±0.30 | **29.68**±0.76 | **32.29**±0.75 | **36.09**±0.74 | **39.82**±0.72 |
| PPO +HYPRL | 1000 | 55.37±3.66 | 106.89±4.96 | 126.25±5.10 | 141.06±5.27 | 153.49±5.35 | 183.51±5.70 |
| DQN | 500 | 832.10±9.93 | 897.54±6.95 | 916.40±5.63 | 917.64±5.55 | 917.89±5.54 | 926.33±5.25 |
| DQN +HYPRL | 500 | **114.60**±2.67 | **115.61**±2.56 | **116.61**±2.52 | **117.61**±2.50 | **148.66**±4.34 | **155.39**±4.58 |
| DQN | 1000 | 893.37±9.67 | 953.25±6.55 | 992.10±2.50 | 992.12±2.55 | 996.05±1.52 | 997.19±1.14 |
| DQN +HYPRL | 1000 | **303.63**±6.64 | **319.09**±6.38 | **335.20**±6.21 | **335.20**±6.21 | **337.21**±6.18 | **339.90**±6.22 |

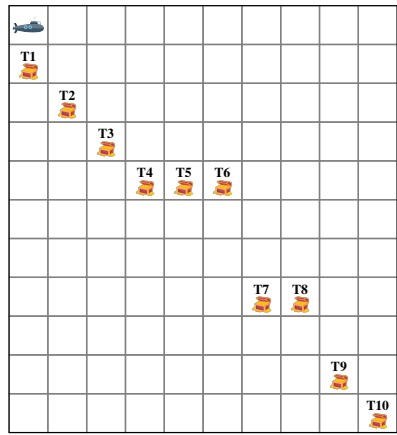

**Figure 9:** DST Grid Map.

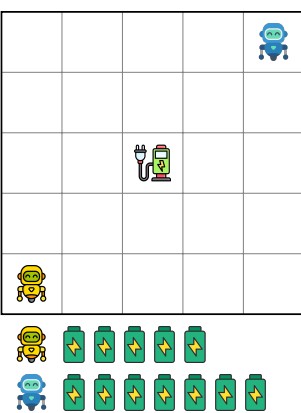

**Figure 10:** Job Scheduling benchmark.

Subsequently, they choose actions from the action space (choosing a domino) to construct traces that aims to satisfy the PCP objective.

Before encoding PCP in HyperLTL, we note that in [19, 9], the authors reduce PCP to the satisfiability problem for the $\forall\exists$ fragment of HyperLTL and the emptiness problem of nondeterministic finite-state hyper automata. Part of the encoding is to ensure that only valid dominos are chosen. We do not need those constraints in our encoding, as the action space of the MDP enforces choosing valid dominos only. This is the reason our that encoding is less complex than that of [19, 9].

We formalize a valid solution to PCP in HyperLTL as follows. The set AP of atomic propositions is defined such that as $\Sigma = 2^{\mathsf{AP}} \cup \{\#\}$, where $\#$ encodes termination. Essentially, the HyperLTL formula requires that for all traces $\tau_1$, where top and bottom words match up to the end of the shorter trace, there exists a trace $\tau_2$ such that $\tau_1$ is a SemiMatch and $\tau_2$ extends $\tau_1$ to complete equal top and bottom words:

$$\varphi_{\mathsf{PCP}} \triangleq \forall \tau_1.\exists\tau_2.\ \psi_{\mathsf{SemiMatch}_{\tau_1}}\ \mathcal{U}\ \left(\psi_{\mathsf{Extend}_{\tau_1,\tau_2}} \wedge \psi_{\mathsf{Match}_{\tau_2}}\right)$$

where $\varphi_{\mathsf{SemiMatch}}$ means the top and bottom words match up to the length of the shorter word:

$$\psi_{\mathsf{SemiMatch}_{\tau_1}} \triangleq \left[\bigwedge_{p\in\mathsf{AP}} \left(p_{top_{\tau_1}} \leftrightarrow p_{bot_{\tau_1}}\right)\right] \mathcal{U}\left(\#_{top_{\tau_1}} \oplus \#_{bot_{\tau_1}}\right)$$

where '$\oplus$' is the xor operator. The formula $\varphi_{\mathsf{Match}}$ indicates that the word constructed on the top and bottom are equal:

$$\psi_{\mathsf{Match}_{\tau_2}} \triangleq \square \bigwedge_{p\in\mathsf{AP}} \left(p_{top_{\tau_2}} \leftrightarrow p_{bot_{\tau_2}}\right)$$

Finally, formula $\varphi_{\mathsf{Extend}_{\tau_1,\tau_2}}$ encodes that trace $\tau_2$ is a successor trace $\tau_1$ as follows:

$$\varphi_{\mathsf{Extend}_{\tau_1,\tau_2}} \triangleq \left[\bigwedge_{p\in\mathsf{AP}} \left(\left(p_{top_{\tau_1}} \leftrightarrow p_{top_{\tau_2}}\right) \wedge \left(p_{bot_{\tau_1}} \leftrightarrow p_{bot_{\tau_2}}\right)\right)\right] \mathcal{U}$$

$$\left(\left(\#_{top_{\tau_1}} \vee \#_{bot_{\tau_1}}\right) \wedge \left(\neg\#_{top_{\tau_2}} \wedge \neg\#_{bot_{\tau_2}}\right)\right)$$

**Experinemt Setup.** We employ DQN as our learning algorithm, utilizing a neural network with three layers of 512 nodes and ReLU activation functions. We set the discount factor $\gamma$ to 0.99, the learning rate to 0.001, and the initial $\epsilon$ to 1.0 with a decay rate of 0.995 down to a minimum of 0.001. We use the Adam optimizer and train for 1000 episodes. This experiment is conducted for domino collections containing 5 and 6 dominos.

**Baselines Reward Functions.** The manually designed reward functions have the following form:

$$R^{\mathsf{PCP}} = \begin{cases} a & \text{if the same indexed letters on the top and bottom are equal,} \\ b & \text{otherwise.} \end{cases}$$

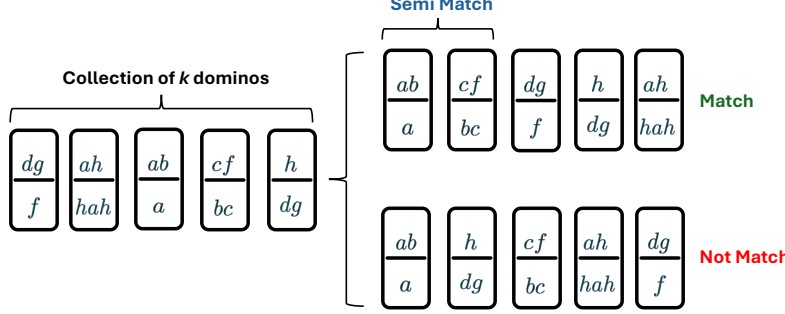

**Figure 11:** PCP overview

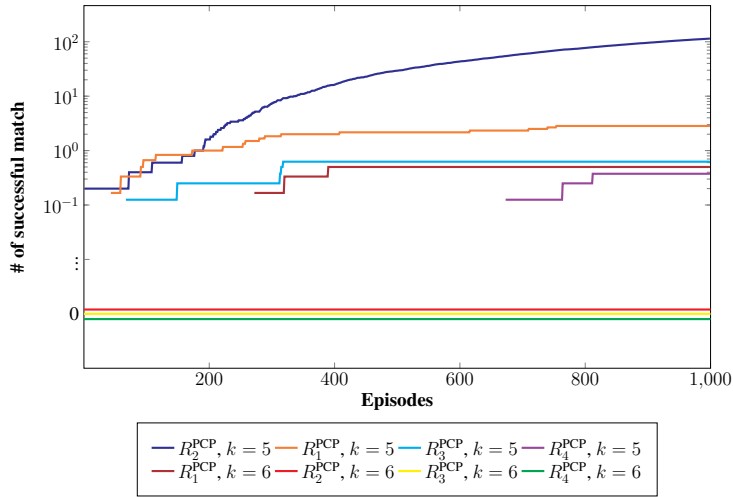

**Figure 12:** PCP baseline results showing average total matches over 10 runs.

The function $R^{\text{PCP}}$ addresses all objectives and safety constraints of the problem. The manually designed reward function is tested with the following values:

- $a = 1, b = -1$ ($R_1^{\text{PCP}}$)
- $a = 5, b = -2$ ($R_2^{\text{PCP}}$)
- $a = 10, b = -10$ ($R_3^{\text{PCP}}$)
- $a = 1, b = -5$ ($R_4^{\text{PCP}}$)

In the paper, we report results using the baseline reward function with values $a = 5$, $b = -2$ ($R_2^{\text{PCP}}$) for $k = 5$, and $a = 1$, $b = -1$ ($R_1^{\text{PCP}}$) for $k = 6$. Figure 12 presents the results corresponding to the different reward functions. The baselines $R_2^{\text{PCP}}$, $R_3^{\text{PCP}}$, and $R_4^{\text{PCP}}$ for $k = 6$ did not successfully produce any matches. We visualize this using a value of zero in the log-scale plot shown in Figure 12.

**Table 7:** Wildfire results, avg. and std. error of 10 trials after 5k episodes.

| Size | Method | Dist | Steps $O_1$ | Steps $O_2$ |
|------|--------|------|-------------|-------------|
| $3^2$ | PPO + $R_1^{\text{FAIR}}$ | 2.8±0.01 | 77.13±5.7 | T/O |
| | PPO + $R_2^{\text{FAIR}}$ | **2.5**±0.01 | **33.43**±4.1 | **787.03**±31.8 |
| $5^2$ | PPO + $R_1^{\text{FAIR}}$ | 5.6±0.02 | **31.2**±4.5 | **7057.8**±1047.9 |
| | PPO + $R_2^{\text{FAIR}}$ | **4.2**±0.01 | 62.7±9.7 | T/O |

### D.4.2 Wild Fire Scenario

**Case Study Setup.** Consider a wildfire scenario in a grid-world environment as we introduced in Section 1. Three locations are on fire, and two victims need to be rescued. Two autonomous agents are deployed from the same position with distinct objectives: the firefighter agent (FF) must extinguish all fire zones, and the medical agent (Med) must rescue all victims. The agents must also satisfy two constraints: which requires them to always remain within a 2-cell communication range; and which prohibits Med from entering any fire zone before FF has extinguished the fire in that zone. The state space is represented as a tuple $\langle x, y \rangle$, and the action space is defined as $a = \{stay, up, down, left, right\}$. The goal is to learn optimal policies for FF and Med that maximize the probability of satisfying all these requirements. The following HyperLTL formula captures the objectives of this case study:

$$\varphi_{\mathsf{Rescue}} \triangleq \forall \tau_1 . \exists \tau_2 . (\psi_{\mathsf{fire}} \wedge \psi_{\mathsf{save}} \wedge \psi_{\mathsf{dist}} \wedge \psi_{\mathsf{safe}})$$

$$O_1: \psi_{\mathsf{fire}} \triangleq \Diamond(i_{\tau_1}) \wedge \Diamond(f_{\tau_1}) \wedge \Diamond(c_{\tau_1}) \qquad C_1: \psi_{\mathsf{dist}} \triangleq \Box(|\mathsf{Location}_{\tau_1} - \mathsf{Location}_{\tau_2}| < 3)$$

$$O_2: \psi_{\mathsf{save}} \triangleq \Diamond(g_{\tau_2}) \wedge \Diamond(f_{\tau_2}) \qquad\qquad C_2: \psi_{\mathsf{safe}} \triangleq (\neg i_{\tau_2} \, \mathcal{U} \, i_{\tau_1}) \wedge (\neg f_{\tau_2} \, \mathcal{U} \, f_{\tau_1}) \wedge (\neg c_{\tau_2} \, \mathcal{U} \, c_{\tau_1})$$

**Experiment Setup.** We use the PPO implementation from [38], with the following hyperparameters: a learning rate of 0.0003, a clipping parameter of 0.2, a discount factor $\gamma = 0.995$, and a GAE lambda of 0.95. We conduct our experiments on $3 \times 3$ and $5 \times 5$ grid-world environments. The timeout is set to 1000 steps for the $3 \times 3$ grid and 10,000 steps for the $5 \times 5$ grid.

**Baselines.** We evaluated (see Table 7) two baseline reward functions in this experiment:

- $R_1^{\mathrm{FAIR}}$: extinguishing fire: $+50$, rescuing a victim: $+10$, agent out of range: $-100$, and Med in a fire zone: $-100$.
- $R_2^{\mathrm{FAIR}}$: Extinguishing fire: $+10$, rescuing a victim: $+50$, agent out of range: $-100$, and Med in a fire zone: $-100$.

We used $R_2^{\mathrm{FAIR}}$ in the paper, although in $5 \times 5$ $R_1^{\mathrm{FAIR}}$ performs better than $R_2^{\mathrm{FAIR}}$ (in terms of only Steps $O_1$ and Steps $O_2$). However, we chose to include $R_2^{\mathrm{FAIR}}$ in the main paper because its performance in the $3 \times 3$ grid is consistently better than $R_1^{\mathrm{FAIR}}$ across all metrics. This trend does not hold in the $5 \times 5$ grid, where $R_1^{\mathrm{FAIR}}$ outperforms $R_2^{\mathrm{FAIR}}$ only in Steps $O_1$ and Steps $O_2$.

### D.5 Additional Experiment

**Case Study Setup.** In this case study (see Figure 10), a single permanent resource is placed on a grid with multiple agents, which must learn to share resources. The objective here is to maximize the overall utility of all agents while ensuring fair allocation of the resource. In other words, the goal is not merely to maximize utility by allocating the resource to a single agent but to distribute it equitably among all agents. The action space is defined as $a = \{stay, up, down, left, right\}$. The state space is extended by $\langle x, y, \mathsf{Energy} \rangle$. We expresses our allocation and fairness objectives by the following HyperLTL formula:

$$\varphi_{\mathsf{FAIR}} \triangleq \forall \tau_1 . \forall \tau_2 . \left( \Box \Diamond \mathsf{Resource}_{\tau_1} \wedge \Box \Diamond \mathsf{Resource}_{\tau_2} \right) \wedge \Box (|\mathsf{Energy}_{\tau_1} - \mathsf{Energy}_{\tau_2}| < \delta)$$

That is, all agents should eventually gain access to the resource at every step, while ensuring that the difference in their Energy levels (i.e., allocated resources) remains less than a threshold $\delta$, which can be set as a hyperparameter[2]. In our setup, agents start with Energy $= 0$, and each time an agent reaches the resource position, its energy level increments by one while maintaining the same action space.

**Experiment Setup.** We employ PPO as our learning algorithm, using two neural networks: a policy network and a value network. Both networks consist of three hidden layers with 512 nodes and ReLU activation functions. The policy network uses a softmax activation in the output layer, while the value network uses a linear output. We set the learning rate to 0.001, the discount factor $\gamma = 0.95$, and the clipping factor to 0.2. Optimization is performed using the Adam optimizer. We also set $\delta = 10$.

---

[2]We acknowledge that $\Box \Diamond$ is a Büchi condition and is strange to be use in finite semantics. Nevertheless, when it is interpreted in the context of robustness values, function $\rho$ attempts to maximize the occurrence of Resource, which is the intended objective.

**Baselines Reward Functions.** The manually designed reward functions have the following form:

$$R^{\text{FAIR}} = \begin{cases} a & \text{if resource allocated to either of agents,} \\ b & \text{if } |\text{Energy}_{\text{Agent 1}} - \text{Energy}_{\text{Agent 2}}| > \delta \end{cases}$$

The function $R^{\text{FAIR}}$ addresses all objectives and safety constraints of the problem. The manually designed reward function is tested with the following values:

- $a = 2, b = -1$ ($R_1^{\text{FAIR}}$)
- $a = 20, b = -5$ ($R_2^{\text{FAIR}}$)
- $a = 10, b = -5$ ($R_3^{\text{FAIR}}$)
- $a = 5, b = -5$ ($R_4^{\text{FAIR}}$)

**Analysis and Results.** The evaluation (see Figure 13) demonstrates how the learning process maximizes utility for both agents while minimizing the difference in their utilities. In Figure 13a, using HYPRL, we observe that the two agents initially start with low resource utilization. Although this issue is addressed over time, there is a noticeable gap in resource allocation between episodes 150 and 270. HYPRL gradually reduces this disparity, and after episode 400, it achieves high resource utilization while maintaining fairness. This is evident from the fact that the gap between the maximum and minimum utilization becomes nearly invisible between episodes 400 and 500. By the end of training, each agent receives approximately 40 to 45 resources per episode (with 100 steps), while the optimal allocation is 50 resources per agent. These results highlight the effectiveness of HYPRL in achieving fairness in MARL.

On the other hand, using PPO with the baseline reward functions does not successfully yield a policy that both maximizes the average utility of the agents and minimizes the gap between the minimum and maximum allocated resources. For example, $R_1^{\text{FAIR}}$ (see Figure 13b) succeeds in maximizing the average resource allocation to the optimal level but fails to minimize the disparity between agents in terms of resource allocation. The other baseline reward functions $R_2^{\text{FAIR}}$, $R_3^{\text{FAIR}}$, and $R_4^{\text{FAIR}}$ exhibit similar behavior (see Figures 13c to 13e, respectively). While they perform better than $R_1^{\text{FAIR}}$ in minimizing the min-max range of resource allocation, they still fall short compared to HYPRL in achieving both fairness and optimal utility.

# E  Why HYPRL is Important?

In this work, we propose a method named HYPRL, which introduces a novel fusion between reinforcement learning and a recently developed class of logics known as hyperproperties. Hyperproperties have gained increasing attention in the formal methods community due to their expressive power in capturing system-level behaviors that span across multiple execution traces. We observe that many objectives and constraints in MARL naturally align with the expressive capabilities of hyperproperties. Motivated by this insight, we present a framework that first formally specifies the objectives and constraints of a given RL problem using hyperproperties and then learns policies that aim to maximize the probability of satisfying these specifications.

We believe that our approach provides a solid theoretical foundation for a new direction in reinforcement learning, where users can formally express their high-level requirements using hyperproperties and then synthesize policies that satisfy them. Importantly, hyperproperties are not limited to specifying objectives of individual agents or pairwise dependencies. Recent extensions such as HyperATL [7] allow reasoning about team-level objectives and inter-team relational dependencies, further expanding the scope of formal specification in MARL.

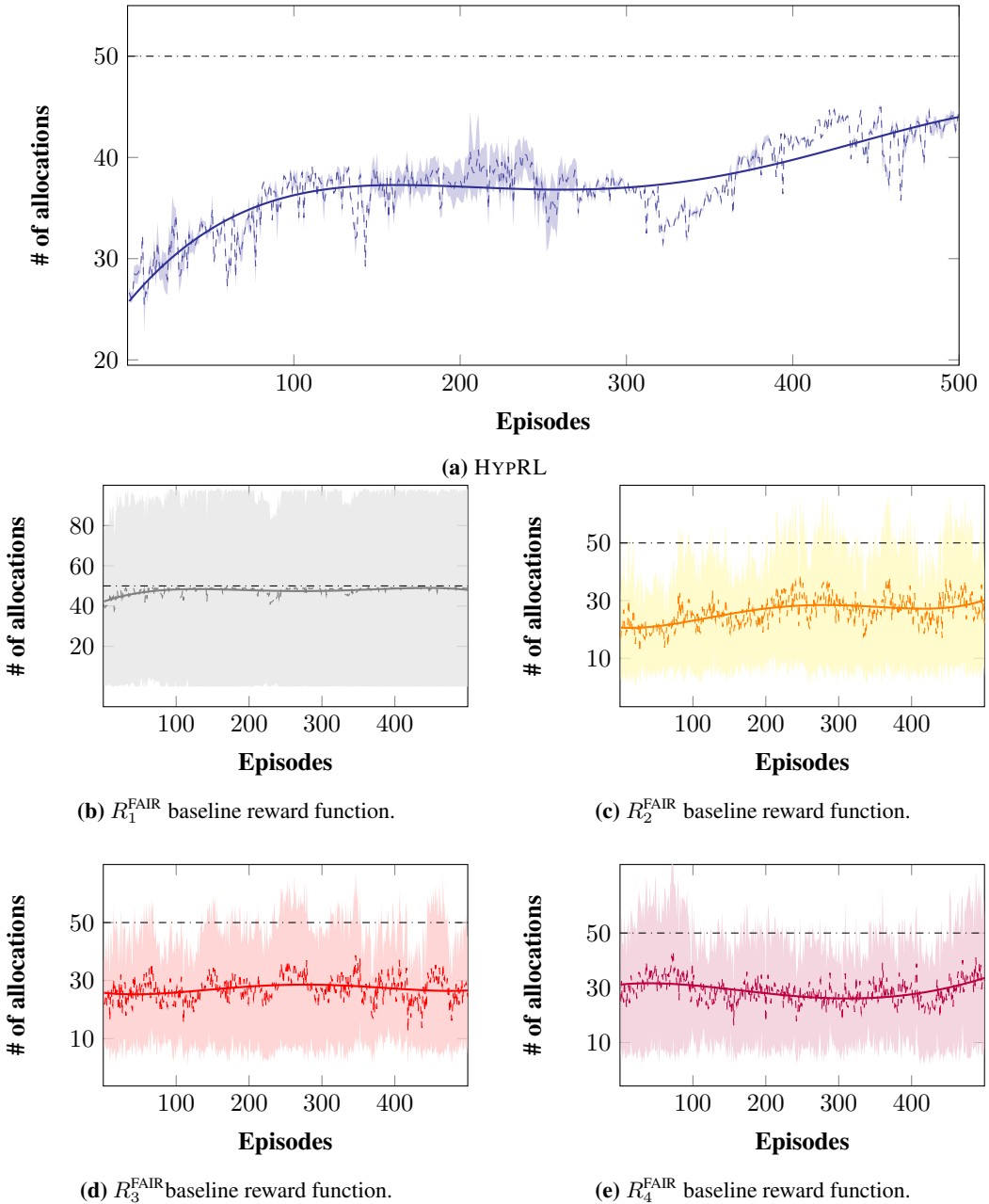

**(a)** HYPRL

**(b)** $R_1^{\text{FAIR}}$ baseline reward function.

**(c)** $R_2^{\text{FAIR}}$ baseline reward function.

**(d)** $R_3^{\text{FAIR}}$ baseline reward function.

**(e)** $R_4^{\text{FAIR}}$ baseline reward function.

**Figure 13:** Fair resource allocation results under various baseline reward functions. Boundaries represent the minimum and maximum allocated resources per episode. Dashed lines indicate average usage, dashed-dotted lines show optimal allocations, and solid lines represent polynomial trend fits. Results are based on 10 independent runs in a $4 \times 4$ grid with two agents, each run consisting of 500 episodes with 100 steps per episode.

