# OpenReview forum: "HypRL: Reinforcement Learning of Control Policies for Hyperproperties"
_NeurIPS.cc/2025/Conference — NeurIPS 2025 poster_

### Official Review · Reviewer_xQCo · 2025-06-27

**Clarity:** 1
**Significance:** 2
**Originality:** 3
**Rating:** 4
**Confidence:** 4

**Summary:**

This paper introduces a novel formal specification guided multi-agent reinforcement learning (MARL) framework where specifications are given as hyperproperties. In particular, they adopt the HyperLTL logic which extends LTL with existential and universal quantifiers over  traces. The authors claim that HyperLTL, in the context of MARL, allows greater flexibility in specifying interdependencies between agent behaviors using quantifier alternations and they demonstrate this with a relevant example. Their method involves first considering a Skolemized version of the specification and using some quantitative semantics of LTL as a reward to train off-the-shelf RL algorithms that gives a centralized policy producing actions for all the agents. They implement their method and test it on three discrete environments.

**Questions:**

My main confusion is about the problem formalization and problem statement. If we have just a single MDP, what does it mean to synthesize multiple policies? How does the quantifier alternation affect optimization?

**Ethical Concerns:**

["NO or VERY MINOR ethics concerns only"]

**Final Justification:**

The original paper had a mistake in formalization of the problem. After extensive back-and-forth with the authors, we trust them to fix the mistake.

**Limitations:**

yes

**Quality:**

2

**Strengths And Weaknesses:**

Strengths
The introduction of hyperproperties in the context of multi-agent reinforcement learning has not been explored before to the best of our knowledge so this is indeed a novel direction.

Weaknesses
Firstly, we believe that the problem statement and the benchmarks used do not show the utility of using hyperproperties. In the current formulation of the problem statement (page 4), it is unclear what is the role of the quantifier alternation because it is trying to maximize the probability that a tuple of traces sampled from the agents' policies together satisfy the specification. In particular, this probability is not dependent on the quantifiers in the specification.

Consider the case of the motivating example used in the introduction. Here, we could define the same specification as an LTL formula over a zipped trace of the firefighter and medical agents and the problem statement would remain the same. This is also the case with all the other benchmark specifications that are considered. In fact, the final policy produced also does not depend on the quantifiers since it is a centralized policy that predicts actions simultaneously for all the agents.

To make this more concrete, if we fix the policies for all the agents, then we can resolve the MDP into a Markov chain for each agent. Then the quantifier-free LTL part of the specifications presented in this paper can be seen as an LTL formula over the product of these Markov chains. So these specifications do not define hyperproperties over this product Markov chain. In other words, interdependence between traces in the Markov chains of each agent specified with an LTL formula is not a hyperproperty. Furthermore, if we did want to consider hyperproperties over the product, it is unclear whether HyperLTL has well defined probabilistic semantics over a Markov chain. We refer to Abraham and Bonakdarpour [1] and Dimitrova et al [2] who define different probabilistic temporal logics to specify hyperproperties over Markov chains and MDPs respectively.

The above problem is accompanied by a lack of a sound and complete formalization. Despite multi-agent reinforcement learning being the focus of the paper, the initial definitions do not specify these multiple agents and how their traces are represented in the MDP. Next, in the definition of the semantics of HyperLTL, it is stated that a tuple of traces $\mathcal{T}$ satisfies the specification $\forall \pi.\phi$, if for all $t\in \mathcal{T}$, $(\mathcal{T}, \Pi[\pi \to t]) \models \phi$, which means that $\phi$ is satisfied by all assignments of traces to $\pi$. However, in the remainder of the paper, the convention seems to arbitrarily shift to being such that the tuple of traces $\mathcal{T}$ represents a sequence of assignments for the quantified variables in the formula. This again seems to point to the fact that the quantifiers play no role.

Finally, the proposed training procedure does not present any novel insights since it uses off-the-shelf RL algorithms with rewards given by quantitative semantics of the LTL formula and only trains a centralized policy that controls all the agents at the same time. Essentially, the multi-agent control problem is turned into a single-agent control problem.

[1] E. Abraham and B. Bonakdarpour. HyperPCTL: A temporal logic for probabilistic hyperproperties. In Quantitative Evaluation of Systems, QEST 2018, Proc., 2018.

[2] Rayna Dimitrova, Bernd Finkbeiner, Hazem Torfah. Probabilistic Hyperproperties of Markov Decision Processes. In ATVA 2020.

---

> ### Author Rebuttal · Authors · 2025-07-30
>
> # Reviewer xQCo
> We deeply thank you for your detailed feedback. We admit we made an unfortunate mistake in formalization of the problem statement, which led to your comments. We are strongly confident that this error can be fixed easily.
> ## Clarifications
> 1. **Probabilities and sampling** First, we clarify a misunderstanding by the reviewer. Please note that in model-free RL, the environment is given by an MDP, where the transitions are unknown to the agents but known to the environment. Since sampling is done by the agent, the transition probabilities only contribute at sampling time (i.e.,transitions with higher probabilities are more likely to appear in sampled traces). After sampling, we have set(s) of traces, and the transition probability function, no longer plays a role in evaluation of the specification or the learning process. This means logics such as HyperPCTL or PHL are irrelevant and not needed.
> 2. **Problem Statement:** Each quantifier is associated to the behavior of an agent. Our goal is to identity a policy per quantifier such that the induced sampled traces are more likely to satisfy the specification. This likelihood (i.e., probability $\mathbb{P}$ in the problem statement) is simply measured by counting the number of cases where sampled traces satisfy the specification divided by the total number of evaluations. Since a policy renders a distribution (i.e., a DTMC), we will easily modify the semantics of HyperLTL to account for choosing traces for each quantifier from the distribution induced by the associated policy.
> Let an *interpretation* be $\mathcal{T} =\langle T_{\tau_i}\rangle_{i \in\lbrace1,\ldots,|\mathit{Vars}(\varphi)|\rbrace}$, where each $\tau_i$ (associated with agent $i$) ranges over a set $T_{\tau_i}$ of traces induced by policy $\pi_i$. This ensures that agent-specific behaviors and quantifier alternations are correctly represented during evaluation. We will mend the semantics of the universal quantifier (analogously for $\exists$) as follows:
> $$(\mathcal{T}, \Pi, 0)\models\forall\tau\,\psi\quad\text{iff}\quad\text{for all }t\in T_{\tau}\text{ such that }(\mathcal{T},\Pi[\tau\rightarrow t],0)\models\psi$$
> Now, consider formula $\varphi=\forall\tau _ 1.\exists\tau _ 2.\psi$, where our goal is to identify policies $\pi^* _ 1$ and $\pi^* _ 2$, such that the sets of sampled traces $T_{\tau_1}$ and $T_{\tau_2}$ from the induced DTMCs (which we denote by "distributions" $\mathcal{D _ {\pi_i}}$), are more likely to satisfy $\varphi$ among all other policies. Formally, $\tau _ i$ for $i \in\lbrace 1,2\rbrace$ ranges over $T _ {\tau _ i} = \mathsf{Traces}(\mathcal{Z} _ {\tau _ i}\sim\mathcal{D} _ {\pi_i})$, where each $\mathcal{Z} _ {\tau _ i}$ is a set of traces sampled from the distribution induced by policy $\pi_i$. The correct formal problem statement will be:
> $$\langle \pi_i^\star\rangle_{i\in\lbrace1,\ldots,n\rbrace}\in
> \Big[ \arg\max _ {\langle\pi_i\rangle}
> \mathbb{P}\left(\langle \mathsf{Traces}(\mathcal{Z} _ {\tau_i}\sim\mathcal{D} _ {\pi_i})\rangle\models\varphi\right)
> \Big] _ {i\in\lbrace1,\ldots,n\rbrace}$$
> Subsequent sections will be revised easily by replacing all instances of $\mathsf{Tr}(\zeta_i)$ by $\mathsf{Traces}(\mathcal{Z} _ {\tau_i}\sim\mathcal{D} _ {\pi_i})$.
> 1. **Illustrative Example:**  We will add the following example to explain all the concepts more clearly using the additional page. Consider the MDP below for a $3\times3$ grid (Fig.1), where each state is $<x,y>$ with $x,y \in\lbrace 0,1,2\rbrace $, labeled by $\lbrace a,b,...,i\rbrace$, and $act=\lbrace0,1,2,3\rbrace$.
> ```
>   <2,0>(a)
>    /  \
>   0    1
>   ↓    ↓  ↘
>   1   .5   .5
>   ↓   ↙      ↘
> <2,1>(b)   <1,0>(d)
> /   \      ↗  | \  ↖
> 3    0    1 ← 0  1   ↖
> ↓    ↓ ↘         ↓     ↖
> 1   .5  .5       1       1
> ↓   ↙     ↘       ↘         ↖
> <1,1>(e) <2,2>(c)  <0,0>(g) - 0
>   |  \     |
>   3   0    1
>   ↓   ↓    ↓
>   1   1    1
>   ↓    ↘   ↓
> <0,1>(h) <1,2>(f)
>   |        |
>   1        3
>   ↓        ↓
>   1        1
>    ↘      ↙
>    <0,2>(i)
> ```
> Suppose agent $A$ with $\pi _ 1$ draws samples $\mathcal{Z} _ {\tau _ 1} =\lbrace\zeta^1_A,\zeta^2_A\rbrace$ from the above MDP (the number of samples can be more than 2):
> $$\zeta_A^1:\underbrace{\langle 2,0\rangle} _ {a}\overset{\text{0}}{\rightarrow}\underbrace{\langle2,1\rangle} _ {b} \overset{\text{0}}{\rightarrow}\underbrace{\langle 2,2\rangle} _ {c}\overset{\text{1}}{\rightarrow}\underbrace{\langle 1,2\rangle} _ {f}\overset{\text{3}}{\rightarrow}\underbrace{\langle0,2\rangle} _ {i}$$
> $$\zeta_A^2:\underbrace{\langle 2,0\rangle} _ {a}\overset{\text{0}}{\rightarrow}\underbrace{\langle2,1\rangle} _ {b}\overset{\text{0}}{\rightarrow}\underbrace{\langle 1,1\rangle} _ {e}\overset{\text{3}}{\rightarrow}\underbrace{\langle 0,1\rangle} _ {h}\overset{\text{1}}{\rightarrow}\underbrace{\langle0,2\rangle} _ {i}$$
> Agent $B$ with policy $\pi _ 2$ draws $\mathcal{Z} _ {\tau _ 2} =\lbrace \zeta^1_B,\zeta^2_B\rbrace$:
> $$\zeta_B^1:\underbrace{\langle 2,0\rangle} _ {a}\overset{\text{1}}{\rightarrow}\underbrace{\langle1,0\rangle} _ {d}\overset{\text{1}}{\rightarrow}\underbrace{\langle 0,0\rangle} _ {g}\overset{\text{0}}{\rightarrow}\underbrace{\langle 1,0\rangle} _ {d}\overset{\text{0}}{\rightarrow}\underbrace{\langle1,0\rangle} _ {d}$$
> $$\zeta_B^2:\underbrace{\langle 2,0\rangle} _ {a}\overset{\text{1}}{\rightarrow}\underbrace{\langle2,1\rangle} _ {b}\overset{\text{3}}{\rightarrow}\underbrace{\langle 1,1\rangle} _ {e} \overset{\text{0}}{\rightarrow}\underbrace{\langle 1,2\rangle} _ {f} \overset{\text{3}}{\rightarrow}\underbrace{\langle0,2\rangle} _ {i}$$
> Now, consider the following HyperLTL formula:
> $$\varphi _ {\mathsf{exp}}\triangleq\forall\tau _ 1.\exists\tau _ 2.\Big(\Diamond i _ {\tau _ 1}\wedge\Box dist(\langle x,y\rangle_ {\tau _ 1},\langle x, y\rangle_ {\tau _ 2})<3\Big)$$
> We calculate the probability of satisfying $\varphi $ using $\mathcal{Z}_{\tau_1}$ and $\mathcal{Z} _ {\tau _ 2}$ (obtained by $\pi _ 1$ and $\pi _ 2$) as follows:
> $$\mathsf{Traces}(\langle\lbrace \zeta_A^1\rbrace,\mathcal{Z} _ {\tau _ 2}\rangle)\models\varphi _ {\mathsf{exp}} ~(\text{where } \zeta_B^2\text{ is a witness to } \exists\tau _ 2)$$
> $$\mathsf{Traces}(\langle\lbrace \zeta_A^2\rbrace,\mathcal{Z} _ {\tau _ 2}\rangle)\models\varphi _ {\mathsf{exp}} ~(\text{where } \zeta_B^2\text{ is a witness to } \exists\tau _ 2)$$
>
> Hence:
> $$\mathbb{P} _ {\langle \pi _ 1, \pi _ 2\rangle}\Big[ \mathsf{Traces}(\langle\mathcal{Z} _ {\tau _ 1},\mathcal{Z} _ {\tau _ 2}\rangle)\models\varphi _ {\mathsf{exp}}\Big]=1$$
> Again, note that, computing the probability in the problem statement ($\mathbb{P}$) is not related to the probability distributions of the MDP.
> Now, if $\varphi _ {\mathsf{exp}}$ was of the form $\forall\forall$, evaluation has to go over all combinations between traces between $\mathcal{Z} _ {\tau _ 1}$ and $\mathcal{Z} _ {\tau _ 2}$, that is:
> $$\mathsf{Traces}(\langle\lbrace\zeta_A^1\rbrace,\lbrace\zeta_B^1\rbrace\rangle)\not\models\varphi _ {\mathsf{exp}}
> ~\mathsf{Traces}(\langle\lbrace\zeta_A^1\rbrace,\lbrace\zeta_B^2\rbrace\rangle)\models\varphi _ {\mathsf{exp}}$$
> $$\mathsf{Traces}(\langle\lbrace\zeta_A^2\rbrace,\lbrace\zeta_B^1\rbrace\rangle)\not\models\varphi_{\mathsf{exp}}
> ~\mathsf{Traces}(\langle\lbrace\zeta_A^2\rbrace,\lbrace\zeta_B^2\rbrace\rangle)\models\varphi_{\mathsf{exp}}$$
> Thus, the probability of satisfaction of $\forall\forall\psi$ would have been 0.5. This example illustrates that the probability of satisfying the HyperLTL formula depends on the form of quantifiers in the given HyperLTL formula.
> ## Response to Detailed Comments and Question
> > [C1] Firstly...Problem Statement...depend on the quantifiers...agents.
>
> > [C2] To make... product of these Markov chains...LTL is not a hyperproperty.
>
> > [C4] In the definition...(T,Π[π→t])⊨ϕ...fact that the quantifiers play no role.
>
> As explained above, after mending the semantics of $\forall$ and $\exists$ to multi-model and the problem statement, quantifier alternation plays a crucial role in evaluation of $\varphi$ and computing $\mathbb{P}$. The product of the induced DTMCs and verifying an LTL formula can only solve the problem for alternation-free formulas.
> > [C3] Furthermore,...[1] and [2]...problem is accompanied...in the MDP.
>
> As noted in the clarification, MDP probabilities are not used in evaluating $\varphi$ and logics such as HyperPCTL[1] or PHL[2] are irrelevant.
> > [C5] Finally, the proposed training procedure does not...single-agent control problem.
>
> Our solution renders a multi-agent solution, but with a centralized controller. We introduce a general reward shaping framework that can be integrated with any off-the-shelf RL algorithm. We do not propose a new RL algorithm and in fact, we do not intend to. Synthesizing a decentralized controller is a whole different line of work. We believe our work is the first step towards hyperproperty-based RL and future papers will study more complex variations. One cannot solve all problems in the first step.
>
> > [Q] Confusion...problem statement...single MDP...? How does the quantifier...?
>
> We again sincerely thank the reviewer for carefully reading the paper and providing constructive feedback. We hope our clarification section addresses the core concerns.
> ## Plans to Resolve the Confusions
> If the paper is accepted, we will obviously fix the mistake and use the additional page to elaborate:
> (1) how the probability of satisfaction of the multi-model semantics HyperLTL formula is calculated using the above example;
> (2) explain the key role of quantifier alternation and how this quantifier alternation affects the calculation of probabilities and consequently affects the optimization problem.
> We believe these minor revisions will resolve the reviewer’s concerns and significantly improve the clarity of the technical meat (Sec. 3–4).
>
> **If our responses have addressed your initial comments and points of confusion, we respectfully ask you to consider raising the score.**

---

> > ### Comment · Reviewer_xQCo · 2025-08-05
> >
> > We thank the authors for their detailed rebuttal and for the clarifications provided. We agree that correcting the mistakes in the semantics and including the example of the calculation of the probability of satisfaction can help readers understand the framework better.
> >
> > It is now more clear how the probability of satisfaction is computed. However, we believe there still remain inconsistencies that require further thought and consideration.
> >
> > Firstly, even though in model-free RL we do not have access to the transition function, we believe it is imperative to have a well-defined probability of satisfaction of a HyperLTL formula in a (known) DTMC.
> >
> > To justify this, let us consider (quantifier-free) LTL specifications. Here, when given a DTMC with full description of its transition probabilities, it is clear how we can define and compute the probability of satisfaction of the LTL specification. This is in fact what probabilistic model checkers do. When we consider LTL specifications in the reinforcement learning setting where we do not know the transition function, a given policy induces a DTMC which we can sample from. The probability of satisfaction of the specification can be approximated by drawing many samples of traces and then counting the number of traces that satisfy the formula. This approximation approaches the true probability of satisfaction with an increasing number of samples. Furthermore, we can show that this convergence is uniform (wrt transition probabilities) for LTLf specification in MDPs with finite horizon.
> >
> > Now consider a HyperLTL specification. In this case, when given DTMCs for all the agents with *known* transition probabilities, it is unclear how to compute the probability of satisfaction of the specification. We believe this should be clarified in a dedicated section, maybe even in the appendix.
> >
> > Here, one might be tempted to define this probability of satisfaction as the limit of the computation presented in the paper when the number of samples grows, but there is a problem. Suppose we have the specification of the form $\forall \tau_1 . \exists \tau_2 . \phi$, and suppose for any behavior of $\tau_1$, there is only one single path in the DTMC induced by $\tau_2$ that can satisfy this formula. If the probability that this single path is taken is very small, say $10^{-5}$, then on average it would take $10^5$ samples to observe this path. Until this path is sampled, the computed probability of satisfaction would be 0 and then suddenly jump to 1. In fact this situation is more problematic in the other direction. If, by chance, we observe this path in the first sample, then even after 1000 samples, we would compute the probability of satisfaction as 1, but in reality the probability that any sampled path satisfies the formula is very small.
> >
> > This problem stems from the semantics of the existential quantifier which only requires one trace to witness the satisfaction of the formula. This is potentially undesirable in the semantics because one could consider a policy $\tau_2$ that *hacks* the satisfaction of the formula by taking the path to satisfy the formula with a very small probability and taking bad/unnecessary actions otherwise.
> >
> > A possible remedy for this could be to pair the existential quantifier with a probability $p$ so that a formula of the form $\forall \tau_1 . \exists_p \tau_2 . \phi$, can be interpreted as: "for all behaviors of $\tau_1$, with probability at least $p$, the joint trace satisfies $\phi$."
> >
> > In conclusion, we do believe hyperproperty-based RL can be a valuable contribution to RL, but we need sound semantics that are both useful as a specification language and tractable for RL.

---

> > > ### Author Response · Authors · 2025-08-05
> > > **Response to Reviewer xQCo (Part 1)**
> > >
> > > We thank the reviewer for their detailed reading and response.
> > >
> > > > It is now ... consideration.
> > >
> > > We are glad the reviewer found the examples in our rebuttal helpful. Regarding the remaining confusion, while we understand and agree with the value of developing formal semantics for HyperLTL for Markov models, we respectfully disagree with the reviewer that such semantics is needed in HYPRL.
> > >
> > > > Firstly, even ... (known) DTMC.
> > >
> > > Indeed, we are in a **model-free** setting, and the evaluation of a HyperLTL formula does not have access to the structure of the induced DTMC. Our revised semantics clearly specifies how to evaluate the formula over sets of sampled traces induced by a given set of policies. This is sufficient and provides the necessary algorithmic detail for our purpose. For example, related work such as[1] also evaluates LTL formulas on sampled traces without relying on DTMC-based model-checking semantics. In addition, defining the suggested semantics is relevant to model checking or model-based RL algorithms.
> > >
> > > We also emphasize that defining HyperLTL semantics for DTMCs **may** introduce a different technique to compute probability $\mathbb{P}$, but it doesn't invalidate our approach. We do not believe our approach bears any incorrectness, wrong, or ill-defined components in the paper. We must clarify that the reviewer’s interpretation of how $\mathbb{P}$ is computed is incorrect. This probability may indeed be 0 or 1 for a **single set of samples**; however, our framework evaluates it over **multiple sets of samples**, which resolves the concern raised.
> > >
> > > > To justify ... (quantifier-free) LTL ...
> > >
> > > > Now consider a HyperLTL ... appendix.
> > >
> > > > Here, one might be tempted ...
> > >
> > > We appreciate the reviewer's thorough thinking and provided these examples. We would like to clarify two points:
> > > 1. Our sample is set-by-set: Given a HyperLTL formula where $|\textit{Vars}(\varphi)|=n$, we **sample $n$ sets in each episode**. For every episode, we evaluate how these $n$ sets satisfy $\varphi$ by calculating the robustness values on that specific sets of $n$.
> > > 2.  In the learning process, the optimal set of policies $\langle \pi_1, \pi_2, \ldots \rangle$ is determined by **which set converges to satisfying the HyperLTL formula with the highest probability**, which includes both:
> > >     - The **probability of satisfying $\varphi$** (which is quantifier-dependent, as we presented in the original rebuttal), and
> > >     - The **probability of drawing such samples** depends on the *unknown transition probabilities* of the induced DTMCs from the MDP. For example, for $n=2$, if in the first episode we observe a high probability of satisfying $\varphi$ using $\langle \pi_1, \pi_2 \rangle$, but in all subsequent episodes we observe a low probability of satisfying $\varphi$ with the same policies, then $\langle \pi_1, \pi_2 \rangle$ will not be considered a "good" policy set for optimizing the probability of satisfying $\varphi$, because the likelihood of encountering such paths in the induced DTMCs is too low.
> > >
> > > To better illustrate this, consider a slightly modified MDP and the two sets of samples below:
> > > ```
> > >   <2,0>(a)
> > >    /  \
> > >   0    1
> > >   ↓    ↓  ↘
> > >   1   .001 .999
> > >   ↓   ↙      ↘
> > > <2,1>(b)   <1,0>(d)
> > > /   \      ↗  | \  ↖
> > > 3    0    1 ← 0  1   ↖
> > > ↓    ↓ ↘         ↓     ↖
> > > 1   .5  .5       1       1
> > > ↓   ↙     ↘       ↘         ↖
> > > <1,1>(e) <2,2>(c)  <0,0>(g) - 0
> > >   |  \     |
> > >   3   0    1
> > >   ↓   ↓    ↓
> > >   1   1    1
> > >   ↓    ↘   ↓
> > > <0,1>(h) <1,2>(f)←↖
> > >   |   ↖     |  \    ↖
> > >   1   0.5← 3   2      ↖
> > >   ↓        ↓   ↓   ↘  ↑
> > >   1       0.5  0.1  0.9
> > >    ↘      ↙    ↓
> > >    <0,2>(i)←←← ↙
> > > ```
> > > ### 1st candidate policies $\langle\pi_1, \pi_2\rangle$: Sampling 1st set of sets of samples $\langle\mathcal{Z}^1 _{\tau _1}$, $\mathcal{Z}^1 _{\tau _2}\rangle$
> > >
> > > Suppose agent A with policy $\pi _1$ draws samples $\mathcal{Z}^1 _{\tau _1} =\lbrace\zeta^1_A,\zeta^2_A, \zeta^3_A\rbrace$ from the MDP:
> > > $$\zeta_A^1:\underbrace{\langle 2,0\rangle}_a\overset{0}{\rightarrow}\underbrace{\langle2,1\rangle}_b\overset{0}{\rightarrow}\underbrace{\langle 2,2\rangle}_c\overset{1}{\rightarrow}\underbrace{\langle 1,2\rangle}_f\overset{2}{\rightarrow}\underbrace{\langle 1,2\rangle}_f$$
> > > $$\zeta_A^2:\underbrace{\langle 2,0\rangle}_a\overset{0}{\rightarrow}\underbrace{\langle2,1\rangle}_b\overset{0}{\rightarrow}\underbrace{\langle 1,1\rangle}_e\overset{3}{\rightarrow}\underbrace{\langle 0,1\rangle}_h\overset{1}{\rightarrow}\underbrace{\langle0,2\rangle}_i$$
> > > $$\zeta_A^3:\underbrace{\langle 2,0\rangle}_a\overset{0}{\rightarrow}\underbrace{\langle2,1\rangle}_b\overset{0}{\rightarrow}\underbrace{\langle 2,2\rangle}_c\overset{1}{\rightarrow}\underbrace{\langle 1,2\rangle}_f\overset{2}{\rightarrow}\underbrace{\langle1,2\rangle}_f$$

---

> > > ### Author Response · Authors · 2025-08-05
> > > **Response to Reviewer xQCo (Part 2)**
> > >
> > > Agent B with $\pi _2$ draws $\mathcal{Z}^1 _{\tau _2} =\lbrace \zeta^1_B,\zeta^2_B, \zeta^3_B\rbrace$:
> > > $$\zeta_B^1:\underbrace{\langle 2,0\rangle}_a\overset{1}{\rightarrow}\underbrace{\langle1,0\rangle}_d\overset{1}{\rightarrow}\underbrace{\langle 0,0\rangle}_g\overset{0}{\rightarrow}\underbrace{\langle 1,0\rangle}_d\overset{0}{\rightarrow}\underbrace{\langle1,0\rangle}_d$$
> > > $$\zeta_B^2:\underbrace{\langle 2,0\rangle}_a\overset{1}{\rightarrow}\underbrace{\langle2,1\rangle}_b\overset{3}{\rightarrow}\underbrace{\langle 1,1\rangle}_e\overset{0}{\rightarrow}\underbrace{\langle 1,2\rangle}_f\overset{3}{\rightarrow}\underbrace{\langle0,2\rangle}_i$$
> > > $$\zeta_B^3:\underbrace{\langle 2,0\rangle}_a\overset{1}{\rightarrow}\underbrace{\langle1,0\rangle}_d\overset{1}{\rightarrow}\underbrace{\langle 0,0\rangle}_g\overset{0}{\rightarrow}\underbrace{\langle 1,0\rangle}_d\overset{0}{\rightarrow}\underbrace{\langle1,0\rangle}_d$$
> > >
> > > So, the probability of satisfying $\varphi_{\mathsf{exp}}$ is 1/3.
> > >
> > > ###  2nd candidate policies $\langle\pi_1',\pi_2\rangle$: Sampling 2nd sets of samples $\langle\mathcal{Z}^2_{\tau _1}$, $\mathcal{Z}^2 _{\tau_2}\rangle$
> > > Agent A with $\pi_1'$ (different from 1st candidate) draws $\mathcal{Z}^2 _{\tau _1} =\lbrace\zeta^1_A,\zeta^2_A, \zeta^3_A\rbrace$:
> > >
> > > $$\zeta_A^1:\underbrace{\langle 2,0\rangle}_a\overset{0}{\rightarrow}\underbrace{\langle2,1\rangle}_b\overset{0}{\rightarrow}\underbrace{\langle 2,2\rangle}_c\overset{1}{\rightarrow}\underbrace{\langle 1,2\rangle}_f\overset{3}{\rightarrow}\underbrace{\langle0,2\rangle}_i$$
> > > $$\zeta_A^2:\underbrace{\langle 2,0\rangle}_a\overset{0}{\rightarrow}\underbrace{\langle2,1\rangle}_b\overset{0}{\rightarrow}\underbrace{\langle 1,1\rangle}_e\overset{3}{\rightarrow}\underbrace{\langle 0,1\rangle}_h\overset{1}{\rightarrow}\underbrace{\langle0,2\rangle}_i$$
> > > $$\zeta_A^3:\underbrace{\langle 2,0\rangle}_a\overset{0}{\rightarrow}\underbrace{\langle2,1\rangle}_b\overset{0}{\rightarrow}\underbrace{\langle 2,2\rangle}_c\overset{1}{\rightarrow}\underbrace{\langle 1,2\rangle}_f\overset{3}{\rightarrow}\underbrace{\langle0,1\rangle}_h$$
> > >
> > > Agent $B$ with $\pi _2$ (same as 1st candidate) draws $\mathcal{Z}^2 _{\tau _2} =\lbrace \zeta^1_B,\zeta^2_B,\zeta^3_B \rbrace$:
> > > $$\zeta_B^1:\underbrace{\langle 2,0\rangle}_a\overset{1}{\rightarrow}\underbrace{\langle1,0\rangle}_d\overset{1}{\rightarrow}\underbrace{\langle 0,0\rangle}_g\overset{0}{\rightarrow}\underbrace{\langle 1,0\rangle}_d\overset{0}{\rightarrow}\underbrace{\langle1,0\rangle}_d$$
> > > $$\zeta_B^2:\underbrace{\langle 2,0\rangle}_a\overset{1}{\rightarrow}\underbrace{\langle1,0\rangle}_d\overset{1}{\rightarrow}\underbrace{\langle 0,0\rangle}_g\overset{0}{\rightarrow}\underbrace{\langle 1,0\rangle}_d\overset{0}{\rightarrow}\underbrace{\langle1,0\rangle}_d$$
> > > $$\zeta_B^3:\underbrace{\langle 2,0\rangle}_a\overset{1}{\rightarrow}\underbrace{\langle 1,0\rangle}_d\overset{1}{\rightarrow}\underbrace{\langle   0,0\rangle}_g\overset{0}{\rightarrow}\underbrace{\langle 1,0\rangle}_d\overset{0}{\rightarrow}\underbrace{\langle 1,0\rangle}_d$$
> > >
> > > So, the probability of satisfying $\varphi_{\mathsf{exp}}$ is 0.
> > >
> > > > This problem ... existential ... otherwise.
> > >
> > > For the 1st sample set, $\langle\mathcal{Z}^1_{\tau_1}, \mathcal{Z}^1_{\tau_2}\rangle$, the probability of satisfaction is $1/3$. However, in $\zeta_B^2$, the transition $\langle 2,0\rangle \overset{1}{\rightarrow} \langle 2,1 \rangle$ has a **probability of only 0.001**, making it extremely unlikely to occur in other samples. Thus, when applying the same “good” policy to $\tau_2$ in the second sample set $\langle\mathcal{Z}^2_{\tau_1}, \mathcal{Z}^2_{\tau_2}\rangle$, the probability drops to 0 since this rare path is unlikely to reappear in the next episode.
> > >
> > > This illustrates that certain policies might incidentally sample an outlier path, thereby skewing the probability of satisfaction. In RL, such outlier paths may occur in only a few sample sets, making their contribution negligible across many episodes. As these rare paths are unlikely to reoccur, the algorithm naturally learns to disregard the corresponding non-optimal policies.
> > >
> > > > A possible...satisfies $\phi$.
> > >
> > > We agree with the reviewer that this alternative may yield valuable results, but our semantics is not incorrect. Even if the proposed semantics produces better numbers in some cases, this does *not imply a flaw in our approach*. RL is inherently a best-effort technique, and our method remains both correct and effective.
> > >
> > > > In conclusion... for RL.
> > >
> > > We appreciate the reviewers’ positive view on incorporating hyperproperties into RL. We are strongly confident that the HyperLTL semantics is sufficient for our **model-free RL framework**, where no formal semantics for HyperLTL over Markov Chains is required.
> > >
> > > **If our responses have addressed your following concerns, we respectfully ask you to consider raising the score.**
> > >
> > > [1]Jothimurugan et al. Compositional reinforcement learning from logical specifications.Neurips21

---

> ### Comment · Reviewer_xQCo · 2025-08-06
>
> Thanks again to the authors for their detailed clarifications. We agree that the presented semantics are sufficient and the algorithm would indeed discard policies that only satisfy the existential operator with low probability. The content presented regarding the semantics in the rebuttals deserves better explanation in the paper. We hope that the authors can incorporate these changes for the final version and we are raising our score.

---

### Official Review · Reviewer_McAK · 2025-06-29

**Clarity:** 2
**Significance:** 2
**Originality:** 3
**Rating:** 5
**Confidence:** 2

**Summary:**

This paper proposes HypRL, a reinforcement learning framework that learns multi-agent control policies guided by specifications written in HyperLTL, a logic for expressing properties over multiple execution traces. Unlike traditional approaches that rely on manually crafted rewards, HypRL derives reward signals from logical robustness values, enabling it to handle complex relational constraints between agents.

The framework uses Skolemization to manage quantifier alternation in HyperLTL and trains policies using standard RL algorithms. Evaluations across tasks like Deep Sea Treasure, and the Post Correspondence Problem show that HypRL significantly outperforms standard RL baselines.

**Questions:**

Do the authors have any intuition or preliminary ideas on how HypRL could be extended to handle partially observable environments?

**Ethical Concerns:**

["NO or VERY MINOR ethics concerns only"]

**Final Justification:**

After considering the author response and discussion, I have increased my score by 1. The authors addressed my main concerns as follows:
- Additional experiments on larger grid sizes demonstrate improved scalability.
- Clarifications show HYPRL improves performance within the same training budget.
- The authors outlined a clear path toward extending HypRL to Dec-POMDPs, which is a promising direction.
- While the paper remains dense, the authors plan to enhance accessibility in the final version.

Overall, the paper presents a novel and technically sound approach.

**Limitations:**

yes

**Quality:**

3

**Strengths And Weaknesses:**

## Strengths
- [S1] The wildfire scenario offers an intuitive and accessible introduction to the problem and why standard reward shaping fails.
- [S2] The method uses HyperLTL and Skolemization to define a novel, theoretically grounded framework for learning multi-agent policies under complex constraints.
- [S3] The paper avoids ad hoc reward engineering by using logical robustness values as a principled reward signal.

## Weaknesses
- [W1] The environments, while varied, are still relatively small in size and abstraction. It would be valuable to evaluate HypRL on more complex domains, if possible.
- [W2] Although HypRL outperforms PPO in the wildfire scenario, the number of steps required remains large compared to the optimal policy, raising concerns about sample efficiency and scalability.
- [W3] The technical content, especially the use of Skolemization and robustness evaluation, is quite dense and may hinder accessibility for a broader audience.

---

> ### Author Rebuttal · Authors · 2025-07-30
>
> # Reviewer McAK
>
>
> Thank you very much for your thoughtful and constructive review. We truly appreciate the time and care you put into reading our paper. Below, we provide detailed responses to each of your questions and concerns.
>
> ## Clarifications
>
> 1.	**Complex Domain:**
> We thank the reviewer for this suggestion. We have added experiments on larger grids ($8 \times 8$ and $10 \times 10$, see details below) for the Wildfire domain to demonstrate HYPRL’s effectiveness in more complex environments. If the paper is accepted, we will ensure these results are clearly highlighted in the final version.
>
> 2.	**Extension to Partially Observable Environments:**
> While this paper focuses on MDPs, the HYPRL framework is naturally extendable to Dec-POMDPs, allowing for reasoning under partial observability. We will be sure to include a discussion of this promising direction in the final version if accepted.
>
>
> ## Response to Detailed Comments
>
> > [W1] The environments, while varied, ... complex domains, if possible.
>
> Yes, we agree that the environments in our experiments are abstracted due to limited computational power. However, we aimed to include a variety of experiments to address this limitation. For example, the SRL environments are standard and widely used benchmarks in the (Safe) RL community (e.g., [1], [2], [3]). Furthermore, cases such as the PCP remain challenging to handle because the underlying problem is **undecidable**. Regarding the wildfire domain, for this rebuttal, we extended the environment to $8 \times 8$ and $10 \times 10$ grids (2 victims and 3 firezones). The table below reports the averages and standard errors over 10 trials, each consisting of 5k episodes, repeated across 10 runs. We set (T/O) to 20,000 steps for the $8 \times 8$ grid experiments and 30,000 steps for the $10 \times 10$ grid experiments, respectively.
>
>
> |Grid| Method    | Dist | Step $ O_1 $    | Step $ O_2 $  |
> | -------- | -------- | ------- | -------- | ------- |
> 8 * 8 | PPO  | 11.27 ± 0.03    | 16801.8 ± 2144.0 | T/O    |
> 8 * 8| PPO + HYPRL | **6.94** ± 0.07    | **4149.6** ± 1743.1 | **386.2** ± 80.5     |
> 10 * 10 | PPO  | 10.90 ± 0.01    | T/O | 29023.6 ± 976.4   |
> 10 * 10| PPO + HYPRL | **5.39** ± 0.06    | **21272.8** ± 3579.0 | **570.3** ± 52.2    |
>
>
> [1] ElSayed-Aly, I., Bharadwaj, S., Amato, C., Ehlers, R., Topcu, U., & Feng, L. (2021). Safe multi-agent reinforcement learning via shielding.
>
> [2] Melo, F. S., & Veloso, M. (2009, May). Learning of coordination: Exploiting sparse interactions in multiagent systems.
>
> [3] Melcer, D., Amato, C., & Tripakis, S. (2022). Shield decentralization for safe multi-agent reinforcement learning.
>
> > [W2] Although HYPRL outperforms PPO in the wildfire scenario ... sample efficiency and scalability.
>
> Thank you for this comment. The purpose of this experiment is to demonstrate that, **under the same number of training episodes, HYPRL+PPO outperforms PPO**. This suggests that, given the same training budget, HYPRL guides the agents to achieve the task objectives quicker. The wildfire scenario is particularly challenging because each agent has multiple objectives that are dependent (i.e., the FireFighter drone must extinguish fire zones, while the medical drone must rescue victims, and a victim may locate in a fire zone). In addition, they must satisfy complex constraints (i.e., maintaining a safe distance, while ensuring that the medical drone does not enter fire zones before the firefighter extinguishes the fires). These requirements make solving this problem challenging and hence, require training over a large number of episodes.
>
> > [W3] The technical content ... is quite dense and may hinder accessibility for a broader audience.
>
> Thank you for this comment. We agree the technical content is dense, but the mathematical rigor is necessary to ensure soundness. To support the reader, we have included additional material in the appendix. If the paper is accepted (which allows an extra page in the camera-ready version), we will use the additional space to provide further explanation and illustrative examples to improve readability.
>
> ## Questions
>
> > [Q1] Do the authors have any intuition or preliminary ideas on how HYPRL could be extended to handle partially observable environments?
>
> We sincerely thank the reviewer for this insightful and forward-looking perspective on HYPRL. We are, in fact, currently working on partial observability. We believe HYPRL offers a strong theoretical foundation with the potential to open an exciting new line of research. Extending the current setting from MDPs to Decentralized Partially Observable Markov Decision Process (i.e., Dec-POMDPs) would not only allow HYPRL to handle partially observable environments but also address scalability challenges inherent to centralized training. As a preliminary direction, we envision replacing full environment states with **belief states** for trace sampling, while retaining HYPRL’s foundations to evaluate those traces and provide meaningful feedback to the learning algorithm.
>
>
>
>
>
> **If our responses have addressed your initial comments, we respectfully ask you to consider raising the score.**

---

> > ### Comment · Reviewer_McAK · 2025-08-05
> >
> > Thank you for the detailed and thoughtful responses. I appreciate the additional experiments on larger grid sizes and the clarification regarding the extension to partially observable settings. I also acknowledge the authors' plans to improve accessibility and presentation in the final version.
> >
> > Given these clarifications and improvements, I am willing to increase my score by 1.

---

> > > ### Author Response · Authors · 2025-08-07
> > > **Response to Reviewer McAK**
> > >
> > > We appreciate the reviewer’s constructive feedback and thank them for raising the score. We will use the additional page in the camera-ready version wisely to improve the clarity and overall reading experience.

---

### Official Review · Reviewer_16CQ · 2025-07-02

**Clarity:** 2
**Significance:** 2
**Originality:** 3
**Rating:** 4
**Confidence:** 3

**Summary:**

This paper introduces HYPRL for learning multi-agent control policies using HyperLTL specifications. The key contributions include using Skolemization to handle quantifier alternation and quantitative semantics for reward shaping. Through experiments, the authors show HYPRL outperforms basedline on safety-aware planning and  the deep sea treasure problem.

**Questions:**

How does the complexity of HYPRL scale with the size of the HyperLTL formula (e.g., number of operators, quantifier alternations) and the number of agents involved? Can you provide any theoretical bounds or empirical insights into the practical limits of the approach?

**Ethical Concerns:**

["NO or VERY MINOR ethics concerns only"]

**Final Justification:**

The author has addressed my concerns of scalability. I have adapted my scores accordingly.

**Limitations:**

Yes

**Quality:**

3

**Strengths And Weaknesses:**

Strengths:
1. The idea of incorporating HyperLTL for expressing complex multi-agent objectives is novel to me. As HyperLTL can capture relational properties and dependencies among agents, the proposed approach has advantage over standard LTL based approaches.
2. The definition of quantitative semantics for HyperLTL, which translate satisfaction checking into an optimization problem, is a clever way to enable reward shaping for RL.
3. This paper provides a thorough empirical evaluation on different benchmarks. The results demonstrate significant improvements.

Weaknesses:
1. While HyperLTL is natural for multi-agent tasks, it is also much more complex compared to other simpler logics, which could be more expensive to solve. The paper could have discussed more about the scalability of the HYPRL approach, both in terms of the number of agents and the complexity of the HyperLTL specifications, and how it compares to alternative approaches.
2. When comparing to other baselines like PPO, the reward function is not carefully designed for your use cases. I believe one can learn/design a more powerful reward function with IRL or imitation learning.

---

> ### Author Rebuttal · Authors · 2025-07-30
>
> # Reviewer 16CQ
>
> Thank you very much for your thoughtful and constructive review. We truly appreciate the time and care you dedicated to reading our paper. Below, we provide clarifications and detailed responses to each of your questions and concerns.
>
> ## Clarifications
>
> 1.	**Comparison with IRL/Imitation Learning:**
> HYPRL operates without expert demonstrations, unlike IRL or imitation learning, which rely on such inputs. That is, HYPRL is specification-guided, while IRL/Imitation learning are demonstration-guided. Since the assumptions and goals differ fundamentally, a direct comparison would be inappropriate. We’ll clarify this distinction in the final version of the paper if it's accepted.
>
> 2.	**Scalability of HyperLTL Reasoning:**
> We appreciate the reviewer’s concern. HYPRL does not involve any logical reasoning. While HyperLTL logical decision procedures such as model checking and synthesis are expensive, the core technical procedure of HYPRL (Skolemization and robustness-based reward shaping) is based on typical numerical computations and is interactable. We’ll emphasize this important point if our paper is accepted.
>
>
> ## Response to Detailed Comments
>
> > [W1] While HyperLTL is natural ... expensive to solve ... scalability of the HYPRL approach ... and how it compares to alternative approaches.
>
> Thank you for this question. We do not expect any significant scalability issues in HYPRL in either the skolemization or computation of robustness functions. HyperLTL is computationally expensive in exhaustive decision procedures, such as model checking and synthesis, but our approach does not require any such decision procedure. Regarding comparable approaches in terms of scalability, to the best of our knowledge, there is no prior work that synthesizes policies to **maximize the probability of satisfying an arbitrary hyperproperty using RL**. Moreover, while there are studies that use temporal logics, they either cannot be applied in MARL settings, or cannot reason about inter-agent dependencies. Please refer to page 2, lines 59–67, and page 3, lines 68–79 for elaborations.
>
> > [W2] When comparing ... IRL or imitation learning.
>
> In both IRL and Imitation Learning (e.g., [1], [2]), one of the inputs to the system is a set of traces performed by expert agents. In HYPRL, we do not assume access to such expert traces. Instead, we use a HyperLTL formula that captures all the system requirements and aim to synthesize optimal policies that sample sets of traces which maximize the probability of satisfying the HyperLTL formula. Therefore, we believe that **comparing IRL or IL with HYPRL is not an apple-to-apple comparison** due to the different system inputs. In addition, in all our experiments, except for DST (where we use the original reward function from citation [35]), we have tried several reward functions that consider the requirements of the problem for fair comparison and report the *best one* as the baseline (please check Section E of the appendix for elaborations on these reward functions).
>
> [1] Lin, Xiaomin, Stephen C. Adams, and Peter A. Beling. "Multi-agent inverse reinforcement learning for certain general-sum stochastic games." Journal of Artificial Intelligence Research 66 (2019)
>
> [2] Yang, Fan, et al. "Bayesian multi-type mean field multi-agent imitation learning." Advances in Neural Information Processing Systems 33 (2020)
>
> ## Questions
>
> > [Q1] How does the complexity ... any theoretical bounds or empirical insights ...?
>
> Thank you for pointing this out. We acknowledge that HYPRL does not provide theoretical bounds or guarantees. In the SRL example, where we have a reachability goal accompanied by a safety constraint, HYPRL improved upon the shielding baseline in terms of steps required to reach the goal by up to 85\% (i.e., 75.2 for CQ+Shield and 11.90 for CQ+HYPRL in the Pentagon case) while also preserving safety in two cases (i.e., SUNY and MIT, CQ+HYPRL both guarantee 0.0 collisions). Our improvements are not limited to planning benchmarks; we also observed significant gains in solving the PCP problem, which is undecidable. In the setting with six dominos, HYPRL drastically outperformed the ad-hoc reward mechanism, solving more than 150 instances within 1,000 episodes, whereas the ad-hoc approach could not solve any. These experiments provide empirical insight into how HYPRL accelerates learning in multi-agent settings with relational dependencies.
>
> ## Additional Experiment
>
> We conducted additional experiments on the Wildfire domain with larger grids to better support reviewer’s concern regarding the scalability of HYPRL and provide more empirical insights (on [Q1]). We believe Wildfire is a challenging problem due to its complex constraints and requirements, and expanding the domain significantly increases the difficulty of the problem, making it a suitable setting to demonstrate scalability.
>
> The table below reports the averages and standard errors over 10 trials, each consisting of 5k episodes, repeated across 10 runs. We set the time-out (T/O) to 20,000 steps for the $8 \times 8$ grid experiments and 30,000 steps for the $10 \times 10$ grid experiments, respectively.
>
> |Grid| Method    | Dist | Step $ O_1 $    | Step $ O_2 $  |
> | -------- | -------- | ------- | -------- | ------- |
> 8 * 8 | PPO  | 11.27 ± 0.03    | 16801.8 ± 2144.0 | T/O    |
> 8 * 8| PPO + HYPRL | **6.94** ± 0.07    | **4149.6** ± 1743.1 | **386.2** ± 80.5     |
> 10 * 10 | PPO  | 10.90 ± 0.01    | T/O | 29023.6 ± 976.4   |
> 10 * 10| PPO + HYPRL | **5.39** ± 0.06    | **21272.8** ± 3579.0 | **570.3** ± 52.2    |
>
>
> **If our responses have addressed your initial comments, we respectfully ask you to consider raising the score.**

---

> ### Author Response · Authors · 2025-08-06
> **Open to Further Discussion (Reviewer 16CQ)**
>
> Thank you for your acknowledgment of our rebuttal. We remain fully willing to engage in further discussion and address any remaining questions or concerns you may have.
>
> We greatly appreciate your time and constructive feedback throughout this process.

---

### Official Review · Reviewer_UWS5 · 2025-07-03

**Clarity:** 2
**Significance:** 3
**Originality:** 3
**Rating:** 4
**Confidence:** 3

**Summary:**

This paper introduces HYPRL, a reinforcement learning framework that solves the difficult problem of designing rewards for complex multi-agent tasks described using hyperproperties. The framework allows users to define intricate objectives and dependencies between agents using the formal logic of HyperLTL which is then automatically translated into a numerical reward signal (i.e., a robustness score). This is  maximized via a centralized reinforcement learning (RL) algorithm to find coordinated policies that satisfy the original requirements.

**Questions:**

1. What are the scalability limits of the approach in terms of the state-space and action-space dimensions studied?
2. How would the system prioritize safety (less collisions) without a manually constructed shield?
3. How are max-min in the robustness semantics differentiated through? Have “soft” versions of max-min been used?
4. What are the exact state-space or action-space augmentations used in the RL algorithm? Is there a way to decentralize execution?

**Ethical Concerns:**

["NO or VERY MINOR ethics concerns only"]

**Final Justification:**

The paper's core contribution, i.e. using hyperproperties to reason over the joint traces of all agents, is a significant and welcome departure from typical multi-agent methods that focus on specifications for individual agents. However, the primary claim of scalability is not yet substantiated. While the case studies are insightful, the validation is limited to a two-agent setting ($N=2$). Consequently, the argument for scalability to the general N-agent case remains tenuous, especially given the inherent complexity of the specifications considered. The concept is promising, but its broader applicability requires more extensive empirical evidence.

**Limitations:**

- Only shown on a 2-agent case (scaling to n-agents may pose challenges)
- The algorithm is centralized making it unsuitable for large agent populations.

**Quality:**

3

**Strengths And Weaknesses:**

## Strengths:

- HyperLTL captures multi-agent tasks not describable easily by vanilla LTL such as quantifier logic dependent on different agents. It solves the tasks by means of Skolemization of the existential quantified agent policy which is then fed to a centralized RL algorithm.
- The reward shaping is automatic from the specification and the approach is shown to be agnostic to the RL algorithm used with applications using PPO, DQN and CQ-Learning.

## Weaknesses:

- Only considers the 2-agent setting and a single universal plus existential quantifier in sequence in the experiments.
- Assumes fully observed environments which may be hard in the presence of multiple other agents.
- It appears that collisions are **more** than the vanilla case of CQ in the SRL environments (albeit in a shorter completion time). Given that safety is part of the specification, this is concerning.

---

> ### Author Rebuttal · Authors · 2025-07-30
>
> # Reviewer UWS5
>
> Thank you very much for your thoughtful and constructive review. We truly appreciate the time and care you put into reading our paper. Below, we provide detailed responses to each of your questions and concerns.
>
> ## Clarifications
>
> 1.	**HYPRL is Not Limited to Two Agents:**
> While our case studies focus on two agents for clarity, the HYPRL framework theoretically supports arbitrary quantifier structures and is likely to empirically scale to larger number of agents. We will make this generality more explicit in the paper.
> 2.	**HYPRL Does Not Require a Shield:**
> Unlike shielding approaches that rely on pre-synthesized safety mechanisms, HYPRL directly encodes relational safety and coordination constraints as HyperLTL specifications. This allows reasoning over complex inter-agent dependencies without the need for external shielding, which is a key strength of HYPRL. That said, HYPRL does not provide safety guarantees during training.
>
> 3.	**Exact Min–Max in the Robustness Function:**
> In our framework, we obtain scalar reward signals using the robustness function; therefore, the gradient flow is independent of the min/max operations performed in the robustness computation. Moreover, we do not use soft min/max approximations for this purpose.
>
> ## Response to Detailed Comments
>
> > [W1] Only considers the 2-agent ....
>
> Thank you for raising this point. While our experiments focus on a two-agent setting with a single $\forall\exists$ quantifier alternation, our framework is not limited to this configuration. The Skolemization technique we introduce is general and can handle **arbitrary numbers of agents and arbitrary quantifier alternations** (e.g., $ \lbrace \forall^* \exists ^ * \rbrace^ * $, $ \lbrace \exists ^* \forall^ * \rbrace ^* $). Importantly, even within this setting, the tasks we address, such as the **undecidable problem** PCP, remain highly nontrivial and demonstrate the capability of our approach to handle complex relational requirements. We also note that our work is truly the first in addressing specification-guided RL for multi-agent settings with inter-agent dependencies and paves the path for future engineering extension with a large number of agents. We also believe these results serve as a strong proof of concept that HYPRL not only introduces a novel way to leverage HyperLTL formulas for reward shaping in theory but also demonstrates **practical improvements** over existing approaches such as manually designed rewards and shielding-based in safe RL benchmark (which we conducted thorough comparisons in our paper).
>
> > [W2] Assumes fully observed environments ....
>
> We sincerely thank the reviewer for highlighting this exciting future direction. We fully agree with your comment. In this work, we began with fully observable MDP-based environments to clearly establish the core contribution of using HyperLTL for reward shaping. Extending HYPRL's strong theoretical framework to partially observable environments, such as Decentralized Partially Observable Markov Decision Process (i.e., Dec-POMDPs), where agents have different observation spaces, is promising. Indeed, we are currently working on this very problem.
>
>
> > [W3] It appears that collisions ....
>
> Thank you for this observation. We would like to clarify that, in **SUNY** and **MIT** (please check Table 3), HYPRL achieves **zero collision** while maintaining shorter completion times. In **ISR**, CQ and CQ+HYPRL perform almost identically regarding safety. The only notable difference is in **PENTAGON**, where CQ+HYPRL shows a slightly higher collision rate compared to CQ. However, HYPRL achieves significantly faster task completion in this case. One can always combine HYPRL with shielding to achieve full safety.
>
> ## Questions
>
> > [Q1] What are the scalability limits of the approach in terms of the state-space and action-space dimensions studied?
>
> We do not expect the Skolemization-based transformation and robustness function to introduce significant scalability bottlenecks. They are general and can handle arbitrary numbers of agents and quantifier alternations without adding complexity beyond the underlying RL training. The scalability challenges arise primarily from centralized training. However, while we pay the cost of centralized learning, our approach also addresses a key challenge in centralized training: **agent laziness** in cooperative tasks, where some agents may contribute less toward satisfying the overall objective.
>
> > [Q2] How would the system prioritize safety (less collisions) without a manually constructed shield?
>
> Thank you for this question. This point highlights one of the key advantages of HYPRL, that is, we do not require **a pre-synthesized shield** to enforce safety. Instead, we express the safety constraints directly in HyperLTL and use HYPRL to shape rewards derived from this specification. This reward shaping guides policy synthesis toward behaviors that preserve safety without the need of an external safety module. More broadly, this approach is not limited to safety; any relational or inter-agent dependency can be captured in HyperLTL and automatically incorporated into the robustness function via HYPRL.
>
> > [Q3] How are max-min in the robustness semantics differentiated through? Have “soft” versions of max-min been used?
>
> We use exact min/max operations. After we apply Skolemization in Section 4.1, we use the robustness function (which takes the Skolemized HyperLTL formula and zipped traces) mentioned in Figure 2 to compute robustness values that serve as scalar reward signals. Therefore, there is no need to differentiate (i.e., allow gradient flow) through the robustness function in our optimization during the learning process.
>
> > [Q4] What are the exact state-space or action-space augmentations used in the RL algorithm? Is there a way to decentralize execution?
>
> To address the first part of the question, we have provided detailed descriptions of the state and action spaces for all experiments in Section E of the appendix, included in the supplementary materials. Due to space limitations, this could not be included in the main submission; however, if the paper is accepted, we will integrate these details into the final version to improve clarity and completeness
>
> To address the second part of the question, we believe this work opens exciting new directions by applying hyperproperties to multi-agent policy synthesis. While our initial focus is on centralized MDPs, we fully agree that extending the framework to decentralized settings, such as Dec-POMDPs, is a natural and important next step. This extension would enable integration with centralized training and decentralized execution (CTDE) algorithms within the synthesis process. We sincerely thank the reviewer for highlighting this valuable perspective, which aligns closely with our ongoing efforts. We are actively exploring this direction and have observed promising early results.
>
> ## Additional Experiment
>
> To better address the reviewer’s concern regarding the scalability of HYPRL (regarding [Q1]), we conducted additional experiments on the Wildfire scenario with larger grids. We believe Wildfire is a challenging problem due to its complex constraints and requirements, and expanding the environment significantly increases the difficulty of the problem, making it a suitable setting to demonstrate scalability. The table below reports the averages and standard errors over 10 trials, each consisting of 5k episodes, repeated across 10 runs. We set the time-out (T/O) to 20,000 steps for the $8 \times 8$ grid experiments and 30,000 steps for the $10 \times 10$ grid experiments, respectively.
>
> |Grid| Method    | Dist | Step $ O_1 $    | Step $ O_2 $  |
> | -------- | -------- | ------- | -------- | ------- |
> 8 * 8 | PPO  | 11.27 ± 0.03    | 16801.8 ± 2144.0 | T/O    |
> 8 * 8| PPO + HYPRL | **6.94** ± 0.07    | **4149.6** ± 1743.1 | **386.2** ± 80.5     |
> 10 * 10 | PPO  | 10.90 ± 0.01    | T/O | 29023.6 ± 976.4   |
> 10 * 10| PPO + HYPRL | **5.39** ± 0.06    | **21272.8** ± 3579.0 | **570.3** ± 52.2    |
>
> **If our responses have addressed your initial comments, we respectfully ask you to consider raising the score.**

---

> > ### Comment · Reviewer_UWS5 · 2025-08-05
> >
> > I appreciate the author’s clarifications and the additional experiment on scaling the state space for the Wildfire experiment.
> >
> > **Scalability:** The experiments with a larger gridsize help alleviate some concerns of scalability but the main issue of scaling to more than 2 agents (in practice) remains. While HypRL is not restricted to this setting, it is curious why adding a single additional agent into the SRL experiments is computationally intensive. As it is, without any experiments with $N>2$ agents, it is hard to claim that the costly quantifier-based specifications can be reasoned over in a general multi-agent setting. Even a demonstration with $N=3$ agents on a simplified version of one of the tasks would significantly strengthen the paper's claims
> >
> > **Collisions in runtime:** The additional collisions in the **ISR** and **Pentagon** maps over the vanilla CQL approach could point to the difficulty in weighing safety and performance objectives especially in the presence of more complex hyperproperties. While using a manually constructed shield could help, this is not trivial in many settings. Ideally the approach would provide a knob to tune or scalar parameter that allows either of these two conflicting forces (safety v. performance) to be prioritized. Nevertheless, vanilla RL approaches are known to be noisy and perhaps the collisions observed are just an artifact of the noisy policies being run with agents in close proximity to each other.

---

> > > ### Author Response · Authors · 2025-08-06
> > > **Response to Reviewer UWS5**
> > >
> > > We thank the reviewer's response and we are glad our additional experiments help alleviate some of the concerns.
> > >
> > > >Scalability
> > >
> > > We conduct the SRL experiment with **three agents ($N = 3$)** on four maps for SafeRL (ISR, Pentagon, SUNY, MIT). The reported results include the average and standard error of the steps required for all agents to reach their goals (step) and the number of collisions (col), measured over 10 evaluation runs after 1k training episodes, averaged across 5 runs.
> > >
> > > Maps|CQ (step)|CQ (col)|CQ+HYPRL (step)|CQ+HYPRL (col)
> > > -- | -- | -- | -- | --
> > > ISR| 98.79±0.86|12.68±3.85|**74.18±5.19**| **7.78±1.02**
> > > Pentagon| 97.15 ± 2.44| 16.46±7.18|**78.82±1.71**|**10.92±1.44**
> > > SUNY| 84.89±7.94| **0.63±0.24**|**44.95±8.32**|0.71±0.43
> > > MIT| 96.96±1.79| 2.83±1.38|**71.53±7.75**|**1.58±0.71**
> > >
> > > These **initial results** serve as proof-of-concept that, even with more agents, HYPRL+CQ outperforms CQ in almost all cases, both step and col, although col in SUNY is slightly higher. This demonstrates that HYPRL can effectively guide RL algorithms when scaling either the state space (as shown in earlier table on $8\times8$ and $10\times10$ grids) or the number of agents (as in the newly added table with $N=3$). Under the same training budget, HYPRL often achieves better performance.
> > >
> > > We emphasize that, due to the tight discussion timeline, we were only able to present results for $N = 3$ agents on this benchmark in response to the reviewer’s concern. The reviewer may also notice higher steps and collisions for HYPRL (compared to the 2-agent case), which are due to the training setup being limited to 1k episodes under the same time constraints. However, we believe these numbers could be further improved with additional training time. Regarding baseline comparison, we were unfortunately unable to include the shielding method [1] in this experiment, as synthesizing shields is computationally intensive, and we were limited in both time and computational resources.
> > >
> > > We hope this additional table addresses the reviewer's concerns on scalability of HYPRL. We plan to include comprehensive experiments with $N=3$ or more agents, with longer training episodes, on both SRL and wildfire benchmarks, and provide detailed discussions of these additional results in the paper if our paper is accepted, to reflect this valuable point raised by the reviewer.
> > >
> > > > Collisions in runtime
> > >
> > > We appreciate the reviewer for this observation. In HYPRL, the prioritization of safety versus performance can be controlled by adjusting the hyperparameter $c$ in the quantitative semantics (Figure. 2). For example, for SRL, different segments of the inner LTL formula represent either safety or performance requirements, as follows:
> > >
> > > $$ \forall \tau _1.\exists \tau _2. \underbrace{\Box\langle x _ { \tau _ 1 }, y _ {\tau _ 1}\rangle\neq\langle x _ {\tau _ 2}, y _ {\tau _ 2}\rangle} _{\text{no collision (safety)}} \wedge
> > > \overbrace{\Diamond\langle x _ {\tau _ 1},y _ {\tau _ 1} \rangle = \langle x _ {G1},y _ {G1} \rangle \wedge  \Diamond \langle x _ {\tau _ 2}, y _ {\tau _ 2} \rangle=\langle x _ {G2},y _ {G2} \rangle}^{\text{reaching goals (performance)}}  $$
> > >
> > > When interpreting the satisfaction, different values of $c$ can be applied to different segments of the inner LTL formula to accommodate the trade-off. For instance, to *emphasize safety*, we first use the Manhattan distance as the function $f$ to interpret the inequality:
> > >
> > > $$\texttt{Dist}(\langle x_{\tau_1},y_{\tau_1} \rangle, \langle x_{\tau_2},y_{\tau_2}\rangle)>c,$$
> > >
> > > which can be rewritten in the semantics as:
> > >
> > > $$- \texttt{Dist}(\langle x_{\tau_1}, y_{\tau_1} \rangle, \langle x_{\tau_2}, y_{\tau_2}\rangle)<-c.$$
> > >
> > > This interpretation enforces that the *distance* between the two agents must be greater than $c$ to avoid collisions, with the robustness value computed as:
> > >
> > > $$-c+ \texttt{Dist}(\langle x_{\tau_1}, y_{\tau_1} \rangle, \langle x_{\tau_2}, y_{\tau_2}\rangle).$$
> > >
> > > The minimum robustness value is $-c$ (when the agents collide). Here, by increasing $c$, collisions yield *more negative robustness values*, thereby placing greater emphasis on safety (compared to the performance segment in the inner LTL formula).
> > >
> > > We thank the reviewer for raising this valuable point regarding the balance between safety and performance. As illustrated above, HYPRL is capable of supporting fine-tuning by adjusting the hyperparameter $c$ for different subformulas within the LTL body. In our current implementation, we use a neutral setting of $c=1$; however, different values can be selected to place greater emphasis on safety (see Appendix E.1 for more details). We will ensure that this clarification on how to adjust $c$ to balance the trade-off between safety and performance, is included in the final version.
> > >
> > > **We welcome any additional questions or concerns during the remaining discussion period and are happy to address them.**
> > >
> > > [1]ElSayed-Aly et al. Safe multi-agent reinforcement learning via shielding.(2021)

---

> ### Comment · Reviewer_UWS5 · 2025-08-07
>
> The additional experiments with $N=3$ agents in the SRL setting are helpful and alleviate (some of) my concerns. Ideally a general scalability limit of the centralized approach would also be studied. I was curious how the final specification was described. More precisely, is there a global specification that applies to all three agents, such as the one shown below?
>
> $$
> \forall \tau_1 . \exists \tau_2 \exists \tau_3. \Box ( \langle x_{\tau_1}, y_{\tau_1} \rangle \neq \langle x_{\tau_2}, y_{\tau_2} \rangle ) \wedge \Box ( \langle x_{\tau_1}, y_{\tau_1} \rangle \neq \langle x_{\tau_3}, y_{\tau_3} \rangle ) \wedge \Box ( \langle x_{\tau_2}, y_{\tau_2} \rangle \neq \langle x_{\tau_3}, y_{\tau_3} \rangle )  \wedge \Diamond \langle x_{\tau_1}, y_{\tau_1} \rangle = \langle x_{G1}, y_{G1} \rangle \wedge \Diamond \langle x_{\tau_2}, y_{\tau_2} \rangle = \langle x_{G2}, y_{G2} \rangle \wedge \Diamond \langle x_{\tau_3}, y_{\tau_3} \rangle = \langle x_{G3}, y_{G3} \rangle
> $$
>
> or is the specification applied to every pair of agents to calculate the reward which is averaged? I expect the former, but this is just a clarification to understand how these specifications would scale when $N>2$ and what quantifier order would be desired. I hope the authors will be able to provide the shielding baseline as well in the final version of the paper or mention a time-out within a reasonable computational budget. Lastly, an ablation study systematically varying $c$ would be useful to empirically validate that this trade-off is graceful and would provide valuable insight into the model's sensitivity. Would this be similar to just increasing the reward penalty for collisions or does the hyper-property framework balance this in a “smarter” way?

---

> ### Author Response · Authors · 2025-08-08
> **Response to Reviewer UWS5**
>
> We thank the reviewer for the valuable insights and thoughtful questions.
>
> >The additional experiments with $N=3$ ...
>
> We are glad that our new experiment was helpful in addressing part of the reviewer's concern. We appreciate and agree that the scalability limitations deserve further elaboration. In our view, this paper is just the beginning of a new line of research, in terms of both theory and engineering efforts. In the final version, we will include an appendix section reporting experiments with a larger number of agents, detailing the computational cost, and highlighting the practical limits of the learning component. These details were omitted from the current submission because our primary aim in this work is to open a new research direction at the intersection of RL and hyperproperties. Specifically, we introduce a theoretically grounded framework for learning multi-agent policies in environments with complex inter-agent and relational dependencies. While our present focus is on fully observable settings, the framework can naturally extend to partially observable environments by integrating established CTDE algorithms, which help mitigate scalability challenges. Again, we view this paper as the very first step.
>
> >I was curious how the final specification was described. More precisely, is there a global specification ... $N>2$ and what quantifier order would be desired.
>
> Yes, when $N=3$,  the formula has the quantifier structure of $\forall \exists \exists$ or $\forall \forall \exists$ depending on the dependency of the behavior of agents. It is important to note that the complexity of a HyperLTL formula is primarily determined by the number of quantifier alternations. In fact, any formula of the form $\forall ^ * \exists ^ *$ can be reduced to $\forall \exists$, and similarly, $\exists ^ * \forall ^ *$ can be reduced to $\exists \forall$, while $ \forall ^ *$ and $ \exists ^ *$ reduce to $\forall$ and $ \exists $, respectively (see [1] for more details). Our framework, which is based on Skolemization and a robustness function, is capable of handling arbitrary quantifier alternations and does not introduce significant scalability issues while expanding the number of quantifiers (i.e., $N$). That said, in domains without inter-agent or relational dependencies, the property often reduces to a purely universal form ($\forall ^ *$), which can be expressed as an LTL property and is generally less expensive to solve. The aim of this work is to introduce a framework that reasons explicitly over inter-agent and relational dependencies.
>
> >I hope the authors will be able to provide the shielding baseline as well in the final version of the paper or mention a time-out within a reasonable computational budget.
>
> We thank the reviewer for this valuable suggestion. Due to the tight discussion timeline, we could not include the shielding baseline here, but we will incorporate and evaluate it in the camera-ready version.
>
> >Lastly, an ablation study systematically varying $c$ would be useful to empirically ... balance this in a “smarter” way?
>
> We agree with the reviewer that, identifying an optimal value for $c$ is important. However, this is beyond the scope of this study. Of course, HyperLTL syntax has implicit meaning of how "important" each segment of the inner LTL is. For example, given two requirements $A$ (safety) and $B$ (goal reaching), the LTL formula $A \mathcal{U} B$ means "safety must be maintained and the goal has to be reached". However, by using *weak until* (i.e., $A \mathcal{W} B$), one can relax the satisfaction of $B$ and specify "maintain safety all the time, and goal reaching does not have to happen". However, this is less quantifiable compared to adjusting the hyperparameter $c$, and it's hard to argue the real trade-off of safety v. performance.
>
> We thank the reviewer for this insightful point. In our framework, $c$ serves as a hyperparameter that reflects the user’s desired priorities. Moreover, since HyperLTL is temporal, our framework penalizes agents not only for entering unsafe zones but also for approaching them, capturing **relational dependencies**. We see this as a promising direction for future work to explore different trade-offs for varying values of $c$ and will include this discussion in the final version.
>
> **Since the score is not visible to us, we would like to confirm whether we have addressed the reviewer’s concerns and to understand the overall feedback. If there are further questions or concerns, we are happy to respond during the remaining discussion period.**
>
> [1]Finkbeiner, B., & Hahn, C. Deciding Hyperproperties (CONCUR'16)

---

### Author Response · Authors · 2025-08-04
**Reminder for Reviewer Engagement**

We would appreciate it if the reviewers could engage in the discussion. All questions and points of confusion have been addressed in our rebuttal. We look forward to any additional comments or further conversation.

---

### Note · Authors · 2025-08-12

We thank all reviewers for their constructive feedback. Here, we summarize the main concerns, along with our responses and plans to address them.
1. **Calculation of $\mathbb{P}$ Corrected:** Reviewer xQCo raised a concern regarding the role of quantifier alternations in calculating $\mathbb{P}$ (the probability of satisfying a HyperLTL formula $\varphi$ in induced distributions). We acknowledged that this confusion stemmed from a fixable error in formalization of the problem statement. In our rebuttal, we corrected the statement and then **provided a concrete example** showing that quantifiers are indeed crucial in computing $\mathbb{P}$, not only in the satisfaction of $\varphi$ but also in the sampling process over multiple sets of traces. Reviewer xQCo then confirmed (Aug 05) that the computation of $\mathbb{P}$ is now clear, and seem to be moving towards recommending acceptance of the paper (Aug 07). In the final version, we will (1) correct the statement and (2) include the new example with extended explanations for better readability as suggested by reviewer xQCo.
2. **HyperLTL Semantics Clarified:** In our second response to reviewer xQCo (Aug 05), we clarified that probabilistic semantics for HyperLTL is not required in HYPRL, since it is a model-free framework where satisfaction is evaluated over multiple sets of samples. We illustrated with a detailed example that sampling $n$ sets per episode captures both the probabilities of (1) satisfying $\varphi$ and (2) drawing such set of samples. Reviewer xQCo acknowledged (Aug 07) that our HyperLTL semantics are sufficient and our **RL algorithm correctly synthesizes policies with the highest probability of satisfaction**. We will refine the explanation of semantics in the final version to clarify this point.
3. **Extra Scalability Demonstration Added:** In our responses to reviewers UWS5, 16CQ, and McAK, we **conducted two new experiments** to confirm that HYPRL scales effectively both w.r.t. (1) the size of state space and (2) the number of agents. This evidence directly addresses concerns about scalability and shows that HYPRL remains robust across increasing problem sizes (and theoretically, arbitrary number of agents). We will add these experiments and expand the scalability discussion in the final version.
4. **In summary**, the reviewers agree that HYPRL is an exciting and original idea and our proposed solution is valuable. They also seem convinced by our plans to address all concerns raised.

---

### Decision · Program_Chairs · 2025-09-17

**Decision:**

Accept (poster)

**Comment:**

The paper proposes a specification-guided MARL framework that uses HyperLTL to express relational objectives across agents, applies Skolemisation to handle quantifier alternation, and converts formula robustness into a scalar reward usable by standard RL under centralised training. Across multiple environments, HYPRL improves learning efficiency and outcomes over baselines. During rebuttal the authors corrected inconsistencies in the formal problem statement, clarified evaluation over sets of sampled joint traces and added evidence on larger tasks and a three-agent setting; these clarifications and results led multiple reviewers to raise their scores.

Overall the paper is not without its problems: claims of scalability are only partly demonstrated (small environments, limited agent counts, and centralised training), the formulation assumes full observability rather than (Dec-)POMDPs, and the original presentation contained semantic inconsistencies that must be corrected. However, it *is* the first to bring hyperproperties to MARL in a practical way, using HyperLTL to provide principled, algorithm-agnostic reward shaping that reduces ad hoc engineering, with consistent improvements under the same training budget and added evidence on larger grids and a three-agent setup.

Given that the paper is the first to propose the framework, I feel that the shortcomings, though present, need not be addressed in this work. I encourage the authors to include the additional results and corrected semantics in the finalised version of the paper.